# Tsg101 chaperone function revealed by HIV-1 assembly inhibitors

Madeleine Strickland [1], Lorna S. Ehrlich[2], Susan Watanabe[2], Mahfuz Khan[3], Marie-Paule Strub[1], Chi-Hao Luan[4], Michael D. Powell[3], Jonathan Leis[5], Nico Tjandra [1] & Carol A. Carter [2]

HIV-1 replication requires Tsg101, a component of cellular endosomal sorting complex required for transport (ESCRT) machinery. Tsg101 possesses an ubiquitin (Ub) E2 variant (UEV) domain with a pocket that can bind PT/SAP motifs and another pocket that can bind Ub. The PTAP motif in the viral structural precursor polyprotein, Gag, allows the recruitment of Tsg101 and other ESCRTs to virus assembly sites where they mediate budding. It is not known how or even whether the UEV Ub binding function contributes to virus production. Here, we report that disruption of UEV Ub binding by commonly used drugs arrests assembly at an early step distinct from the late stage involving PTAP binding disruption. NMR reveals that the drugs form a covalent adduct near the Ub-binding pocket leading to the disruption of Ub, but not PTAP binding. We conclude that the Ub-binding pocket has a chaperone function involved in bud initiation.

[1] Laboratory of Molecular Biophysics, Biochemistry and Biophysics Center, National Heart, Lung and Blood Institute, National Institutes of Health, Bethesda, MD 20892, USA. [2] Department of Molecular Genetics & Microbiology, School of Medicine, Stony Brook University, Stony Brook, NY 11794-5222, USA. [3] Department of Microbiology and Immunology, Morehouse School of Medicine, Atlanta, GA 30310, USA. [4] High Throughput Analysis Laboratory and Department of Molecular Biosciences, Northwestern University, Evanston, IL 60208, USA. [5] Department of Microbiology and Immunology, Feinberg School of Medicine, Northwestern University, Chicago, IL 60611, USA. Madeleine Strickland and Lorna S. Ehrlich contributed equally to this work. Correspondence and requests for materials should be addressed to N.T. (email: tjandran@nhlbi.nih.gov) or to C.A.C. (email: carol.carter@stonybrook.edu)

The human immunodeficiency virus type-1 (HIV-1) is dependent on the cellular protein, Tsg101, for budding. The recruitment and delivery of Tsg101 to viral assembly sites is accomplished through its interaction with the virally encoded structural precursor polyprotein, group-specific antigen (Gag), which directs viral particle release and has a Pro-Thr-Ala-Pro (PTAP) motif in its C-terminal p6 region that serves as docking site for Tsg101[1–3]. The critical dependence on Tsg101 for productive viral replication is reflected in the fact that, despite motif duplication and extensive genetic heterogeneity in the HIV genome sequence, HIV variants with mutations within the PTAP motif have not been identified to date[4]. The Tsg101 protein is a component of ESCRT-I, one of four complexes (ESCRT 0, -I, -II, -III) that comprise the highly conserved ESCRT (endosomal sorting complex required for transport) machinery. As such, Tsg101 participates in ESCRT-mediated endosomal sorting and trafficking of ubiquitinated cargo to degradative compartments in the cell interior[5, 6]. Gag and the recruited Tsg101 most likely meet in the cytosol and the complex brought to sites of assembly on the plasma membrane by virtue of membrane-binding determinants in the matrix domain of Gag[7, 8].

Central to Tsg101 participation in Gag assembly is its ubiquitin E2 variant (UEV) domain. UEV proteins, and the UEV domain in Tsg101, lack the critical Cys residue essential for conjugation and transfer of Ub to protein substrates or Ub-ligating (E3) enzymes[9, 10]. UEV proteins are highly conserved in evolution and constitute a family of proteins structurally related to, but distinct from, E2 enzymes. The Tsg101 UEV domain contains, in addition to an Ub-binding pocket, another pocket with affinity for PT/SAP motifs[11–14]. We speculated that Tsg101 probably uses its UEV domain to regulate protein levels of other proteins[3]. Our earlier finding that HIV-1 Gag binds Tsg101 through the UEV domain suggested that Tsg101 was recruited as a chaperone to block non-productive Gag ubiquitination that might lead to its degradation, an idea supported by the fact that cyclin-specific E2 enzymes with Ser substituted for the active site Cys are, in fact, dominant-negative inhibitors of cyclin destruction[15]. Mak, Cohen and collaborators demonstrated that, in concert with the E3 ligase MDM2, Tsg101 regulates protein levels of the transcription factor p53[16, 17]. This function was suggested to be independent of the PTAP-binding pocket. Despite indications that Ub plays a critical role in both budding and virus maturation[18–22], how the Tsg101 Ub-binding pocket participates in the virus assembly pathway is currently not known.

Here we provide evidence that the UEV domain of Tsg101 provides chaperone function to HIV-1 Gag that is independent of its interaction with the PTAP motif, supporting the hypothesis that the domain provides a function in addition to its well-established role in ESCRT factor recruitment. Key tools in these studies are agents identified by high-throughput screening of a small molecule library for compounds capable of binding the Tsg101 UEV domain. The inhibitory effects of these probes on

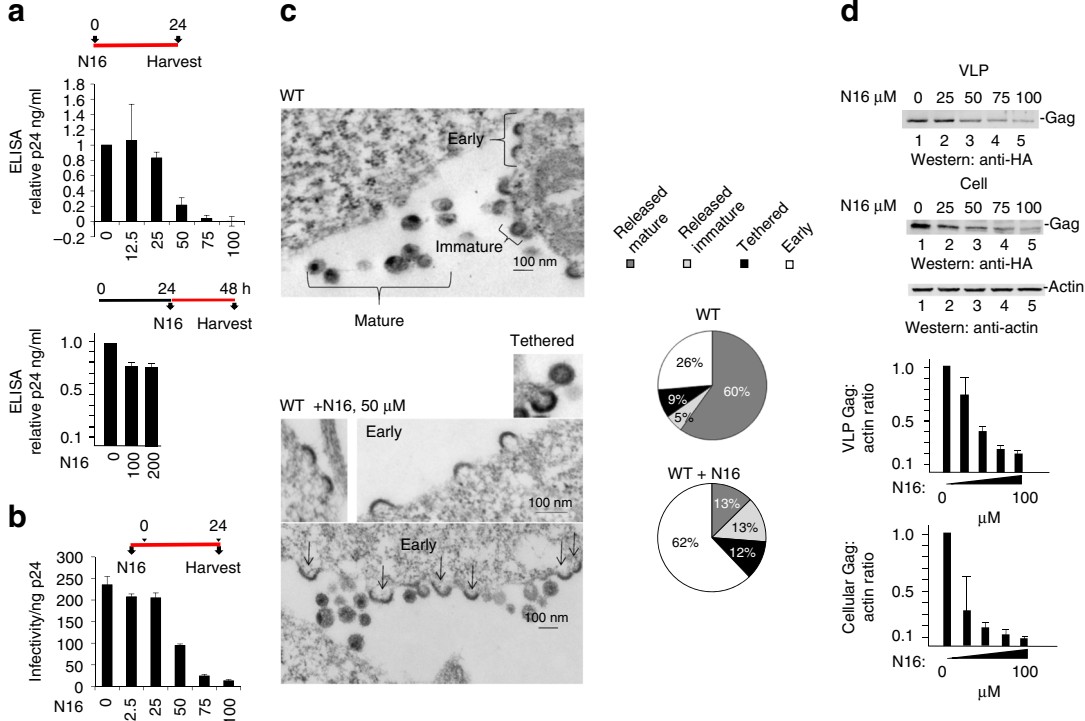

**Fig. 1** N16 inhibits HIV-1 NL4-3 and Gag VLP production. **a** Effect on virus production. Virus particles from N16-treated 293T cells transfected with pNL4-3 as determined by ELISA assay. Top: The compound (μM concentration) was added 6 h prior to DNA transfection (pNL4-3 ΔEnv + HIV-1 IIIB-Env) and tissue culture media was collected 24 h later. Bottom: the compound was added at the indicated concentration (μM) 24 h after DNA transfection and tissue culture media was collected 24 h later. Assay values were normalized to that of the DMSO carrier control. **b** Effect on virus infectivity. Infectious viral particles per ng p24 as determined by MAGI assay. **c** Left: Electron microscopy of particles at cell surface at 24 h post-N16 addition; right: quantitative analysis of budding morphologies. Cells expressing WT NL4-3 were exposed to DMSO carrier or 50 μM N16. Change in 'Early' bud detection: $\chi^2$ p-value < 0.0001. **d** Effects on Gag VLP production. Top: western blot analysis. Cells treated at the indicated concentration were transfected with DNA encoding Gag-HA. At the end of the treatment period, tissue culture media was removed for VLP isolation. Cells were suspended in lysis buffer. Blots were probed for Gag-HA and actin. Bottom: Quantitative analysis of ratio of VLP- or cell-associated Gag to actin normalized to the mock-treated control (0 μM N16). Number of independent trials (n) for **a**, **b**, **c**, and **d** = 3, 2, 2 and 3, respectively; error bars represent 1 SD for the assay values (normalized to 0 μM N16) for **a** and **d**; for **b**, error bars represent 1 SD for the number of infected cells per ng p24

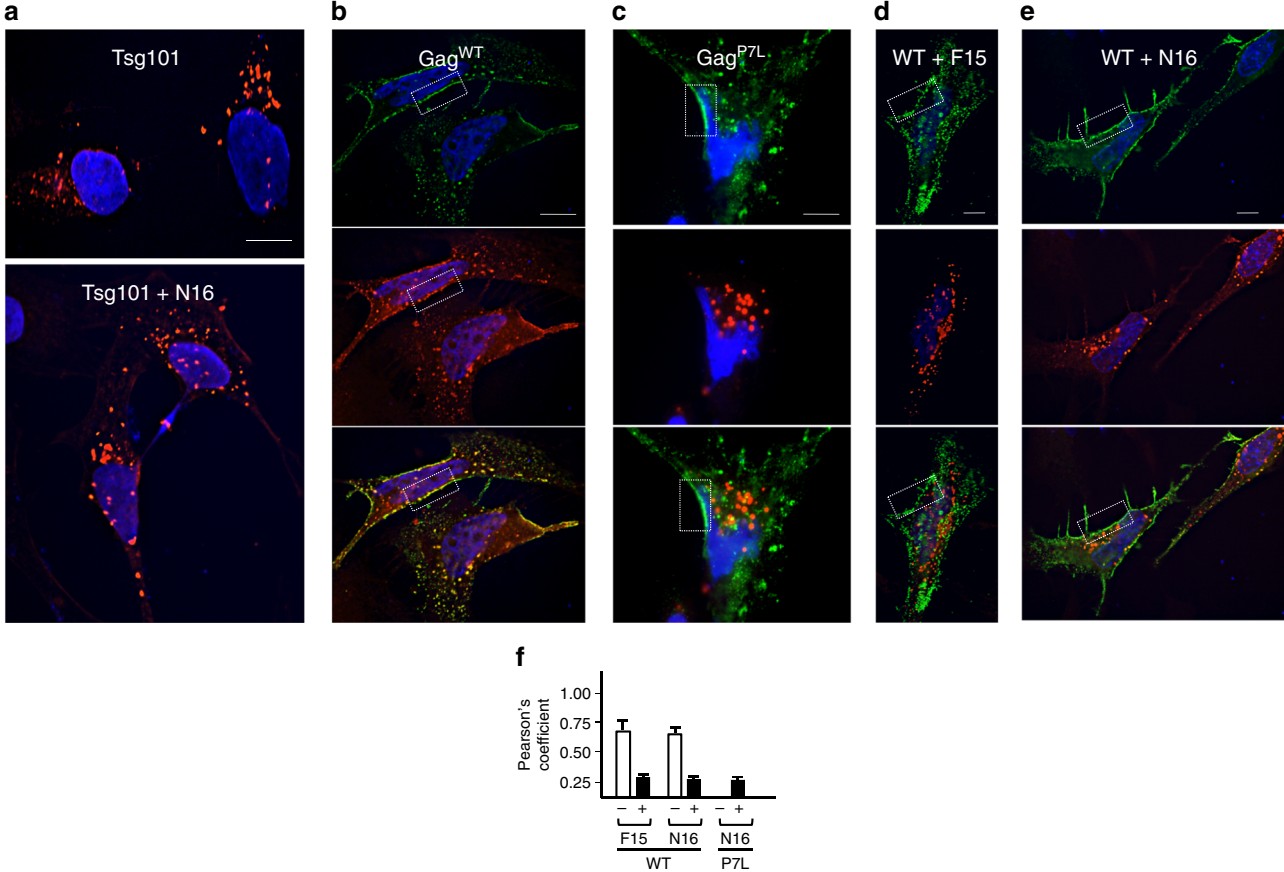

**Fig. 2** N16 (50 µM) prevents co-localization of Gag and Tsg101 on the plasma membrane. **a–e** Fluorescence microscopy images of cells that co-express Tsg101-Myc alone or Gag^WT-GFP and Tsg101-Myc. Cells grown on coverslips were co-transfected with DNA encoding Tsg101-Myc **a** or Tsg101-Myc and Gag^WT-GFP **b**, **d**, **e** or Gag^P7L-GFP **c** and treated with F15 **d** or N16 **e**. After 24 h the coverslips were processed for microscopy with Tsg101-Myc detected with anti-Myc antibody and Texas Red secondary antibody as described in Methods section. Scale bars = 10 microns. **a** Tsg101-Myc in cells treated with DMSO carrier (top) or N16 (bottom). **b–e** Top panels, show signal from Gag-GFP (green); middle panels, images showing Tsg101-Myc signal; bottom panels, merged images showing signals from Gag-GFP and Tsg101-Myc. Boxes frame a region of the plasma membrane. **f** Pearson's coefficient of correlation values determined for the images shown. Error bars indicate the highest values obtained for similar samples. n = 6

HIV-1 Gag assembly which are highly specific suggests that the agents can serve as leads for identification of potent inhibitors of HIV and other pathogens that require Tsg101 participation in viral replication.

## Results

**N16 inhibition of infectious HIV-1 production.** F15 (esomeprazole) and N16 (tenatoprazole) are two related compounds identified in a screen of small molecules capable of binding to the UEV domain of Tsg101 (Supplementary Fig. 1). F15 (trade name Nexium) is widely used for indications of heartburn (also known as acid indigestion). N16 was undergoing phase I clinical trials in July 2016. The compounds share the same heteroaromatic core structure, differing in only one atom, and behaved very similarly in all experiments. To investigate whether the ability to bind the UEV domain could affect Tsg101's function during HIV's budding process, we tested viral particle production (Fig. 1). Addition of N16 6 h before the time of transfection with a plasmid containing the HIV genome (pNL4-3) resulted in a dose-dependent reduction in viral particle production from 293T cells, as indicated by the enzyme-linked immunosorbent assay (ELISA), a test that measures the concentration of the viral capsid (CA) p24 antigen in the cell culture medium (EC$_{50}$ between 25 and 50 µM, Fig. 1a, top). Addition of N16 at 24 h post-transfection had little

effect on budding even at higher concentrations (Fig. 1a, bottom). The latter suggested that the drugs were well-tolerated by cells and that the target of N16 inhibition is an early assembly event. The specific infectivity of the virus was diminished in dose-dependent fashion (Fig. 1b) although higher concentrations were required. N16 also reduced viral particle production in a spreading infection of Jurkat cells infected with NL4-3 (Supplementary Fig. 2). After 15 days, virus production was reduced by 15-fold. Cell viability was maintained under these conditions of sustained drug exposure as indicated by trypan blue viability assay every third day. Subsequent incubation in media without inhibitor resulted in a 10-fold resurgence of the virus after 4 days. Following the re-addition of N16 to the media, virus titer at this point was reduced 50-fold after 4 days indicating maintenance of drug susceptibility.

**N16 arrest of Gag budding.** Electron microscopy was used to identify the event arrested by N16 treatment. Four morphologically identifiable stages comprise HIV-1 viral particle assembly: Deposition of electron-dense material at the cell surface is followed by progressive membrane deformation that results in a protruding bud that is eventually released as an immature virus particle. This particle undergoes morphogenetic rearrangement into the mature infectious particle (Fig. 1c, top, designated as

'Early', 'Tethered', 'Released Immature', and 'Released Mature', respectively). The untreated virus exhibited all of these stages to various degrees. N16 significantly increased representation of the 'Early' stage, thereby impairing production of the mature particle. Given that PTAP mutation, as well as Tsg101 depletion, arrest budding at the 'Tethered' stage, an intuitive expectation is for the N16 arrest phenotype to be the same. However, there is no *a priori* basis for such an expectation as, under conditions of Tsg101 depletion or PTAP mutation, budding is driven by a redundant pathway directed by ALG-interacting protein 1 (AIP1/Alix)[23–27]. As we will show below, N16 disrupts the Ub-binding pocket but leaves the PTAP-binding pocket unperturbed; hence, under conditions of N16 treatment, budding is still Tsg101-PTAP-driven. It is well-recognized that these two budding pathways differ in terms of other cellular proteins engaged[28] and the structure of their bud products[29]. As expected, interference with the Gag[PTAP] Late (L) domain interaction with Tsg101 by substitution of Leu for Pro7 (P7L) in the PTAP motif resulted in accumulation of mainly immature-appearing particles either 'Tethered' to the cell periphery or to each other in addition to the 'Early' form[1, 30, 31]. As indicated by western analysis, treatment of P7L with N16 blocked the residual particle release that the mutation permits (mediated by Alix, an ESCRT adaptor[32]) and EM analysis indicated that the predominant form shifted from 'Tethered' to 'Early' (Supplementary Fig. 3). Collectively, the effect of N16 on WT (Fig. 1) and on the redundant pathway that mediates budding following Tsg101 depletion or PTAP mutation (Supplementary Fig. 3) suggests that the compound targets an event that occurs early in both assembly pathways and necessary for budding progression.

Since Tsg101 involvement in virus production is specific to trafficking and release of the assembled Gag polyprotein precursors, we sought to understand the mechanism underlying inhibition by investigating the effect on Gag function. Gag contains all determinants necessary for formation and budding of immature virus-like particles (VLPs)[7, 8]. As shown in Fig. 1d, N16 treatment was accompanied by dose-dependent reduction in both the steady-state level of Gag intracellular accumulation and the amount of VLP formation. Interestingly, however, although N16 also inhibited VLP release from HeLa cells, Gag intracellular accumulation was not affected (Supplementary Fig. 4a). 293T cells treated with proteasome-specific inhibitor bortezomib[33] throughout the period of N16 exposure failed to stabilize the cytoplasmic Gag (Supplementary Fig. 4b). However, 'ex cellulo' treatment of cells (i.e., treatment just prior to cell lysate preparation) with a mixture of 25 μM MG132[33] (another proteasome inhibitor); and 10 μM N-ethylmaleimide (NEM; an inhibitor of the fusion of late endosomes and lysosomes)[34] preserved Gag in the lysate (Supplementary Fig. 4c). NEM alone was not effective. A mixture of protease inhibitors (aprotinin, 2 μg ml$^{-1}$; pepstatin, 1 μg ml$^{-1}$; leupeptin, 2.5 μg ml$^{-1}$; TPCK, 90 μM; and PMSF, 35 μg ml$^{-1}$) was also effective. Calculation of VLP release efficiency from Supplementary Fig. 4c showed dose-dependent reduction which is consistent with the early budding arrest revealed by EM (Fig. 1c) and categorizes N16 as a budding inhibitor.

At steady-state, endogenous Tsg101 is detected in the cytosol and on the surface of endolysosomal vesicles[35, 36]. Western blot analysis revealed little change in partitioning of the protein to soluble (S1, S3) and particulate (P2, P3) subcellular fractions in the presence or absence of N16 (Supplementary Fig. 5). Fluorescence microscopy was used to examine the effect of N16 on recruitment of Tsg101 by Gag to the plasma membrane. Previously shown aberrant enlarged (>200 nm in diameter) endosomal compartments induced by adventitious expression of the Tsg101 protein (or several other ESCRT subunits)[37, 38] were

seen by fluorescence microscopy of cells expressing Tsg101 tagged with Myc (Fig. 2a, top). N16-induced no apparent change in the size or location of these structures (panel a, bottom). HIV-1 Gag[WT]-GFP (Fig. 2b, top) co-localized with Myc-tagged Tsg101 (Fig. 2b, middle) in these structures and in smaller puncta on the plasma membrane (Fig. 2b, bottom). The Gag[P7L] mutant localized to the cell periphery (Fig. 2c, top, boxed region), however, as expected since its PTAP motif is disrupted, co-localization with Tsg101 (Fig. 2c, middle) was not detected (Fig. 2c, bottom). Under these conditions, Tsg101-Myc was detected as large puncta (~200–300 nm) in the cell interior. To identify the N16-sensitive event, we determined the effect of the compounds following treatment. Like the PTAP mutation in Gag, F15 (Fig. 2d) and N16 (Fig. 2e) did not interfere with the accumulation of Gag on the plasma membrane (top panels); however, they prevented Gag-Tsg101 co-localization at this site (bottom panels) indicating that they both blocked Tsg101 accumulation at the plasma membrane. With both F15 and N16, Tsg101-Myc was detected as large puncta in the cell interior (Fig. 2d, middle and Fig. 2e, middle) and, although significantly less overall Gag-Tsg101 co-localized puncta were detected, the ones detected were detected exclusively in the cell interior. Thus, the affected Tsg101 function was not Tsg101-Gag association per se. A comparison of Pearson's coefficient of correlation values[39] for Gag-Tsg101 co-localization in the presence or absence of N16 is shown in Fig. 2f: Treatment with F15 or N16 reduced the intensity of the co-localization signal by ~twofold. We conclude that the F15 and N16 agents interfere with a Tsg101 function that determines Gag's ability to stably recruit the protein to virus assembly sites on the plasma membrane.

**N16 specificity to antiviral effect**. We investigated the specificity of the inhibitory effect. F15 and N16 impaired Tsg101 co-localization with Gag on the plasma membrane (Fig. 2) and reduced release of the VLPs assembled by Gag (Fig. 1). Under the same conditions (50 μM, 24 h exposure), the N16 compound did not induce detectable cytotoxicity nor did it interfere with the Tsg101 steady-state level, with well-established cell-specified Tsg101 functions such as Tsg101 localization to the midbody of cells undergoing the abscission stage of cytokinesis[40] nor with ligand-induced epidermal growth factor receptor (EGFR) down-regulation[41] (Supplementary Fig. 6; F15 was not tested). Tsg101 localization to the midbody of dividing cells requires its recruitment to that location by centrosomal protein of 55k (CEP55) which binds a site in the Tsg101 Pro-rich domain (aa 154–166)[42]. Finding this function unaffected at a concentration inhibitory to virus budding is consistent with the lack of any indication that cell division was reduced in the presence of N16. The lack of effect on this function also supports the specificity of N16 targeting as limited to functions of the Tsg101 UEV domain. EGF ligand-binding to EGFR on the cell surface signals receptor internalization, ubiquitination, and sequential ESCRT 0-, I-, II- and III-mediated transport to degradative compartments, ultimately lowering the EGFR steady-state level[43]. This trafficking requires the engagement of the P(T/S)AP-binding pocket in the Tsg101 UEV domain with the PSAP motif in hepatocyte growth factor-regulated tyrosine kinase substrate (Hrs), a component of ESCRT-0[41]. The down-regulation function remained unimpaired at concentrations well above 50 μM. The results indicated that these well-established cell-directed Tsg101 functions were resistant to N16 at the concentration to which virus production was susceptible. In contrast, the steady-state level of unliganded EGFR (-EGF lanes) appeared to be N16 sensitive (compare DMSO control/lane 1 and +50 μM N16/lane 3). Intriguingly, constitutive recycling of unliganded EGFR is a recent addition to the list of

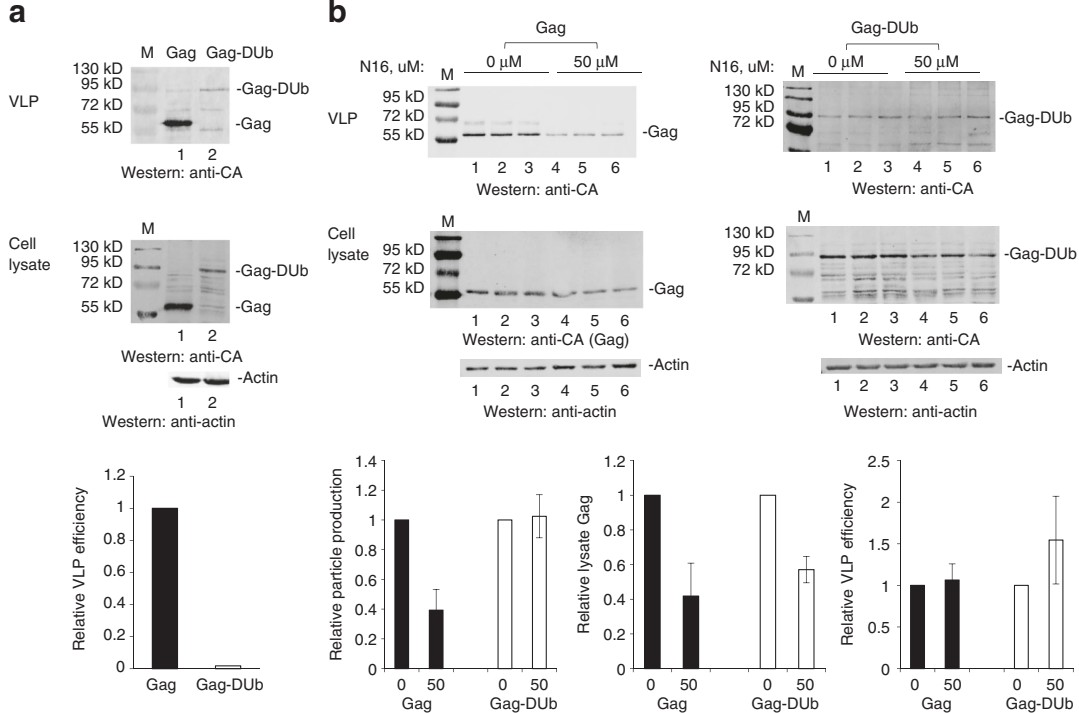

**Fig. 3** Gag-DUb fusion budding is resistant to N16 inhibitory effect. **a** 293T cells were transfected with DNA encoding Gag or Gag-DUb. After 24 h, tissue culture media was removed for VLP isolation and cell lysates were prepared. Top: western blot analysis of Gag in cushioned VLP samples; middle: Gag and actin in cell lysates; bottom: quantitative analysis of VLP release efficiency normalized to the WT Gag control (Gag and Gag-DUb differed in sample size in order to yield visible signal). **b** Effect of N16 on Gag and Gag$^{DUb}$ VLP production. Top: western blot analysis; Triplicate samples are shown to demonstrate reproducibility. Bottom: Quantitative analysis of VLP production (left), lysate Gag (center) and VLP release efficiency (right) normalized to the mock-treated control. Number of independent assays: $n = 3$; error bars represent 1 SD for the assay values normalized to 0 μM N16

cellular functions of Tsg101: Depletion of Tsg101 caused unliganded EGFR to traffic to lysosomes[44].

**Budding of Gag-DUb unaffected by N16**. The fact that fusion of Ub to Gag increases budding efficiency has been reported widely[18, 45, 46]. In contrast, Sette et al.[22] observed that fusion of a deubiquitinating enzyme (DUb) to Gag decreased budding efficiency. We confirmed this observation (see below, Fig. 3a). Together these observations indicate that Ub and DUb contribute to budding in opposing manners. It has also been widely reported that mutation of the PTAP motif in HIV-1 Gag leads to increased Gag ubiquitination[21, 47, 48]. This observation suggests that Tsg101 binding to Gag might be accompanied by recruitment of a Ub peptidase that removes Ub from Gag. Previous studies indicated that Vps23, the orthologue of Tsg101 in yeast, and Doa4, a deubiquitinating enzyme, were both required for removal of Ub from endosomal cargo prior to initiation of multivesicular body (MVB) formation[49]. As budding from the plasma membrane is considered to be topologically equivalent to MVB formation, it seems possible that the observed increased accumulation of early budding structures following N16 treatment (Fig. 1c) was a manifestation of disrupted Ub dynamics during the budding process. If opposing functions of Ub and DUb are both required for budding the imbalance caused by DUb fusion to Gag, whose negative impact on budding was shown previously[22] and recapitulated in Fig. 3a, might be compensated by the N16 targeting of Tsg101.

To test this hypothesis, the effect of N16 on Gag fused to DUb was examined in the absence and presence of N16 (Fig. 3). Figure 3a shows that fusion of the catalytic domain of the Herpes Simplex virus UL36 deubiquitinating enzyme (DUb) onto Gag

inhibited budding, as expected based on previous studies[22]. Quantitative analysis indicated that VLP release efficiency was reduced ~50-fold (Fig. 3a, bottom). Panel 3b shows the effect of N16. As expected based on the results above, budding directed by WT Gag was inhibited by 50 μM N16 (Fig. 3b, top left). In contrast, budding of the DUb fusion was not inhibited when N16 was present (Fig. 3b, top right). Also interestingly, although the N16-imposed vulnerability of the intracellular Gag was unchanged (Fig. 3b, bottom), the overall VLP release efficiency was increased (Fig. 3b, bottom right). The results clearly indicate that the inhibitory effect of N16 treatment can be opposed by DUb, suggesting that the drug targets a previously unappreciated Tsg101 function linked to Ub dynamics during the budding process.

**N16 disruption of Ub binding to the Tsg101 UEV domain**. We solved the high-resolution structure of the N16-Tsg101 UEV complex using NMR, allowing us to probe the atomic interactions between N16 and Tsg101 (PDB ID 5VKG, BMRB ID 30285, Table 1). The N16 binding site of Tsg101 comprised a region surrounding residue C73 that included D40, S41, Y42, N54, T56, W75, and K90 (Fig. 4a, b), which correlated well with the observed chemical shift perturbations of N16 with Tsg101 (Fig. 4c, d, Supplementary Fig. 7). The interaction was stabilized by aromatic π-stacking between the imidazopyridine ring of N16 with residues Y42 and W75 of Tsg101, hydrophobic interactions with T56, and hydrogen bonding between aryl methoxy groups of N16, the backbone NH of S41 and the side-chain amine of K90 (Fig. 4b). Tsg101 chemical shift perturbations caused by PTAP binding showed the same general profile with and without preincubation of N16, with some differences around residues 87–98

### Table 1 NMR structure refinement statistics

| | Tsg101-N16 |
|---|---|
| *NMR distance and dihedral restraints* | |
| Distance restraints | 4961 |
| Total NOE | 4871 |
| Intra-residue | 2201 |
| Inter-residue | 2463 |
| Sequential ($|i-j| = 1$) | 960 |
| Medium-range ($2 \leq |i-j| \leq 5$) | 673 |
| Long-range ($|i-j| > 5$) | 808 |
| Intermolecular | 22 |
| Ambiguous | 207 |
| Hydrogen bonds | 90 |
| Total dihedral angle restraints | 276 |
| $\phi$ | 138 |
| $\psi$ | 138 |
| *Cis*-prolines | P81, P120 |
| *Structure statistics (ensemble)* | |
| Violations (mean and s.d.) | |
| Distance constraints (>0.5 Å) | $14.0 \pm 0.9$ |
| Dihedral angle constraints (>5°) | $0.1 \pm 0.3$ |
| Max. dihedral angle violation (°) | $3.82 \pm 1.29$ |
| Max. distance constraint violation (Å) | $0.53 \pm 0.04$ |
| Deviations from idealized geometry | |
| Bond lengths (Å) | $0.008 \pm 0.000$ |
| Bond angles (°) | $0.852 \pm 0.011$ |
| Impropers (°) | $0.622 \pm 0.016$ |
| Average pairwise r.m.s. deviation* (Å) (all residues) | |
| Heavy | 0.9 |
| Backbone | 0.5 |

NMR distance and dihedral restraints, and structural statistics describing the NMR structural ensemble of the Tsg101-N16 complex. *Cis*-proline residues are also shown. The statistics report on an ensemble of the 20 lowest energy structures from a calculation of 100. *Calculated using the Protein Structure Validation Suite web server[78]

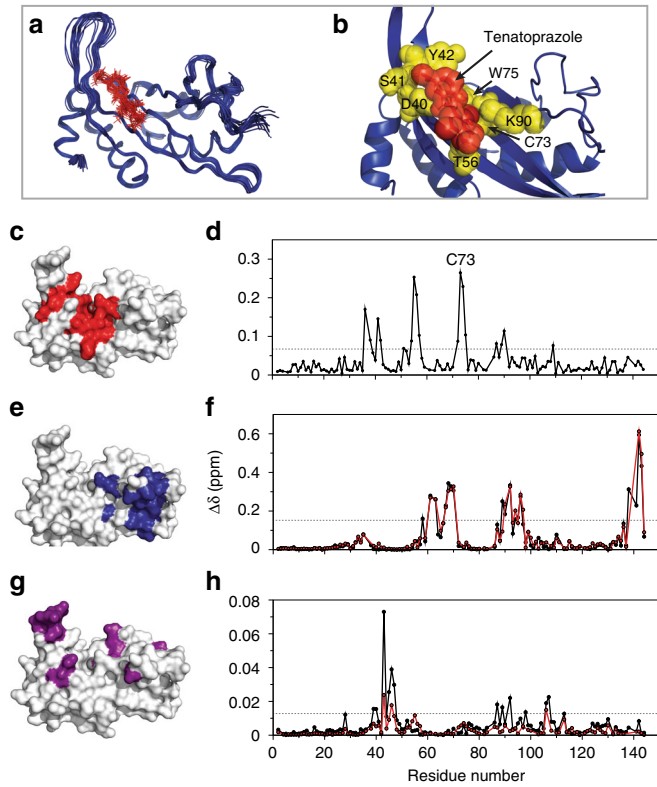

**Fig. 4** Solution NMR reveals that N16 interferes with Ub binding to Tsg101 UEV domain. **a** 100 structures of the N16-Tsg101 UEV complex were calculated of which the twenty lowest in energy are shown in blue backbone trace, with N16 in red lines. **b** Enlarged depiction of the lowest energy structure of Tsg101 UEV (blue) with N16 (red) and binding site residues (yellow) shown as spheres and sticks. NMR restraints and structural statistics are described in Table 1. **c**, **e**, **g** Structure of Tsg101 UEV domain using a surface representation in white (PDB ID: 1KPP)[12] with large chemical shift perturbations (>1.5 standard deviations from zero) upon addition of N16 shown in red **c**, Gag PTAP peptide shown in blue **e**, and ubiquitin shown in purple **g**. **d**, **f**, **h** Residue-specific chemical shift perturbations of Tsg101 UEV following incubation with N16 **d**, PTAP **f**, Ub **h**, all shown with black circles and lines. Large chemical shift perturbations are above the dotted gray line (>1.5 standard deviations from zero). Red circles and lines in **f** and **h** represent pre-incubation with N16 before addition of PTAP or Ub, respectively

(Fig. 4e, f, Supplementary Fig. 8). This contrasts with the Ub-binding pocket perturbation profile, which indicated N16 interfered throughout the pocket resulting in a significantly lower Tsg101 Ub binding affinity in the presence of N16 (Fig. 4g, h, Supplementary Fig. 9).

N16 (Fig. 5a, i) is a prodrug that is acid-activated into derivatives (Fig. 5a, ii and iii) that form disulfide linkages (e.g., Fig. 5a, iv)[50]. The prodrug, but not the charged sulfenamide derivative, can cross the plasma membrane barrier. We speculated that C73, the Cys residue in the Tsg101 UEV domain that was perturbed by N16, formed a disulfide linkage with a derivative produced inside the cell following N16 uptake. We performed mass spectrometry to examine for covalent bond formation following mixing of drug and the Tsg101 UEV domain. This analysis revealed that the interaction between N16 and UEV was covalent, consistent with prodrug conversion to the active sulfenamide form under the conditions of the experiment. The specificity of the effect was confirmed by demonstrating that the addition of N-acetyl cysteine, an antioxidative reagent, prevented the N16 effect (Supplementary Fig. 10). NMR spectroscopy indicated that the β-carbon of Tsg101 residue C73, but not C87, was shifted after N16 binding from ~30 p.p.m. to ~45 p.p.m. as is typical of covalent binding (Fig. 5b). If the sulfenamide derivative of N16 serves as the antiviral agent in the cellular milieu, the substitution of Ala for C73, but not C87, is predicted to permit Gag-Tsg101 co-localization in the presence of the drug by preventing the covalent blockade. As shown in Fig. 5c, d, N16 failed to prevent co-localization of Gag and Tsg101$^{C73A}$-Myc. However, N16 effectively eliminated co-localization of Gag and

Tsg101$^{C87A}$-Myc. Moreover, if Tsg101 residue C73 is the target of N16, then knocking down endogenous Tsg101 and providing a siRNA-resistant version of the C73A mutant should render HIV-1 assembly/release insensitive to N16. This prediction was supported by resistance of budding to N16 treatment in cells in which C73A replaced the endogenous Tsg101 protein. This contrasted with the N16 sensitivity of budding in cells where endogenous Tsg101 was replaced with a siRNA-resistant version of the WT protein (Supplementary Fig. 11). These findings provide further support for the conclusion that Tsg101 is the target of N16 in living cells. Moreover, they demonstrate that C73, the NMR-postulated target of the active drug form, is one of the residues essential for the antiviral effect.

## Discussion

In this study, we identified a class of drugs through a screen (Supplementary Fig. 1) and showed by NMR analysis (i) that its activated form covalently binds the Cys73 residue of the UEV

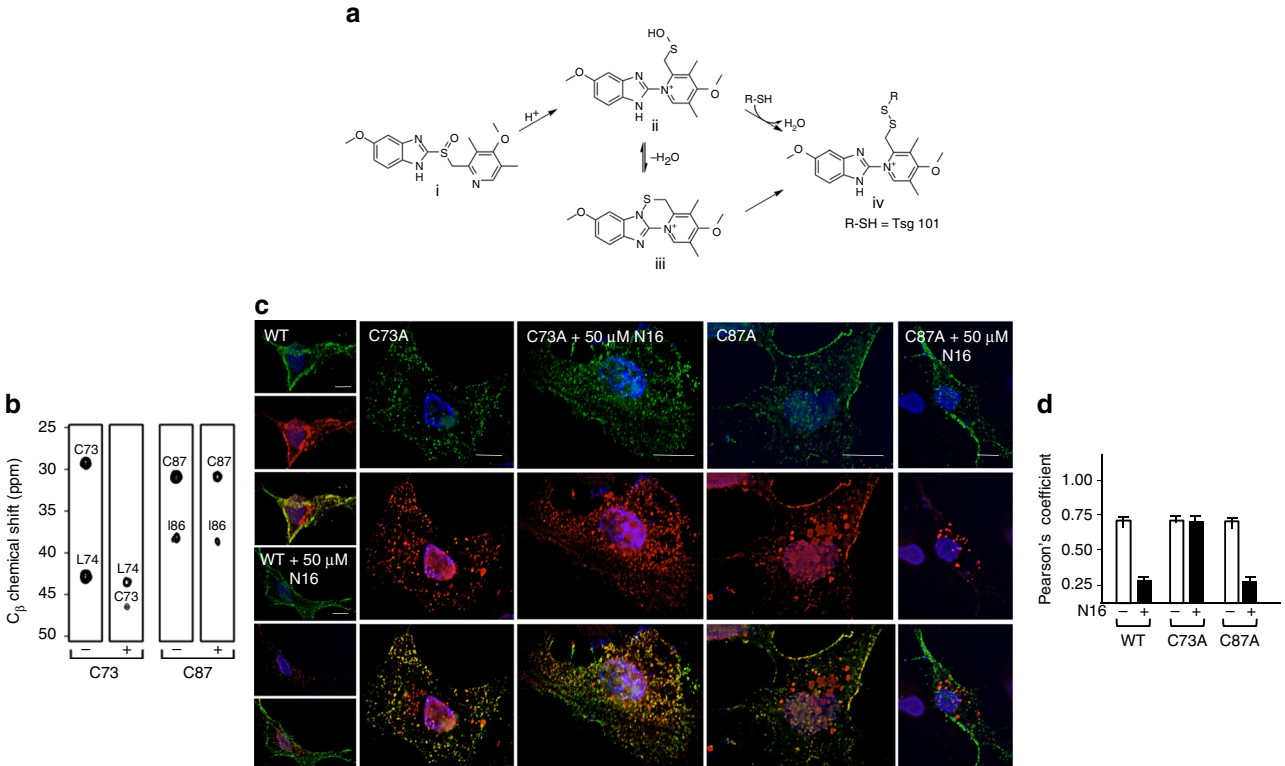

**Fig. 5** N16 binds covalently to C73 in the Tsg101 UEV domain. **a** Activation of N16 to reactive sulfenamide. N16 (i) is converted to intermediates sulfenic acid (ii) and sulfenamide (iii) through acid catalysis to yield a covalently attached N16-Tsg101 UEV complex (iv). **b** HNCACB $C_\beta$ regions for residues C73, L74, I86, and C87 of Tsg101 without (−) and with (+) N16 (1:1 ratio with Tsg101 UEV). **c** Examination by fluorescence microscopy of cells co-expressing Gag^WT-GFP and Tsg101^WT-Myc, Tsg101^C73A-Myc or Tsg101^C87A-Myc in the absence and presence of 50 μM N16. Scale bars = 10 microns. n = 3. **d** Pearson's coefficient of correlation values; error bars represent 1 SD for the coefficient values

domain of Tsg101 and (ii) that this resulted in disruption of UEV Ub binding but not PTAP binding activity (Figs. 4, 5, Supplementary Figs. 7–9). As inhibitors of HIV-1 assembly, the drugs' action appears to be specifically targeted: (1) Inhibitory effects on Gag assembly were observed at concentrations well below drug toxicity as assessed using tests based on cell metabolic activity (Supplementary Fig. 6a). (2) Inhibition of virus production by the drug was neutralized by N-Acetyl Cysteine (Supplementary Fig. 10) supporting the specificity of drug targeting to a Cys residue in the Tsg101-UEV domain (Fig. 5). (3) Inhibition of virus production was suppressed by replacing endogenous Tsg101 with Tsg101 C73A, a Tsg101 mutant lacking the residue with which the drug forms a covalent adduct. In contrast, replacement with WT Tsg101 did not suppress the inhibition (Supplementary Fig. 11). (4) Inhibition of virus production (Fig. 1) and perturbations in trafficking (Fig. 2) and bud formation (Fig. 1) were manifested only when the drug was administered early (i.e., between 6 h pre- and 5 h post-transfection of DNA encoding Gag) and examined 24 h later. When added at 24 h post-transfection and examined 24 h later, Gag assembly was not perturbed even though the drug concentration and exposure time were the same. Thus, based on their specificity for targeting HIV-1 assembly and lack of cellular toxicity, the drugs are useful tools for investigating the contribution of the Ub binding activity of Tsg101 in its various biological roles.

The known role of Tsg101 in HIV-1 production is as conduit to the membrane-remodeling machinery associated with ESCRT-III, which is required for virus budding (reviewed in refs. [28, 51, 52]). The Tsg101-Gag^PTAP binding activity is mainly responsible for the virus' ability to recruit Tsg101 to Gag assembly sites on the plasma membrane. Determinants within Gag direct the complex to budding sites on the plasma membrane where the Tsg101-mediated recruitment of ESCRT-III membrane scission machinery facilitates virus particle release from the cell. Our findings described here reveal that the productive interaction of Gag with the Ub-binding pocket in Tsg101 is another critical contribution of the Tsg101 protein, as it facilitates productive sorting of the protein and progression through the budding stages to the step where ESCRT-III directs ultimate egress. Here or in the cytosol, Gag ubiquitination permits engagement of the Tsg101 Ub-binding pocket, which has been suggested to significantly increase Gag-Tsg101 binding affinity[12]. We speculate that, following UEV binding of F15 and N16 compounds, changes that disturb the Tsg101 Ub-binding pocket prevent Gag-Ub-Tsg101 interaction and plasma membrane localization of the Gag-Tsg101 complex, resulting in the apparently delayed evagination of Gag assemblages on the plasma membrane. However, the notion that promoting high-affinity Gag-Tsg101 interaction is the sole contribution of the Ub-binding pocket to budding does not adequately explain why the known inhibitory effects of disrupting the Gag-Tsg101 interaction in various ways (Tsg101 depletion[1], PTAP mutation[30, 53, 54], dominant-negative Tsg101 expression[37, 54, 55]) result in effects (progression to the stage of tethered particles, defective maturational p25 to p24 processing, polyubiquitination and single released particles arrested in the immature state) that are clearly distinct from effects of exposure to F15 and N16 (aggravated accumulation of predominantly 'Early' buds). This difference indicates that, in addition to any impact on Gag-Tsg101 binding affinity, the Ub-binding pocket in Tsg101 also mediates a function that is critical to the progression of bud evagination. We suggest that this function is L domain-

independent and required whether egress is directed by the PTAP L domain in WT Gag or by the redundant budding pathway that mediates egress under conditions of PTAP mutation or Tsg101 depletion.

Regarding the Gag instability imposed by N16 treatment of 293T cells, it is of interest that treatments to Tsg101 (depletion followed by replacement with C73A) or Gag (fusion of DUb to Gag) were capable of rescuing the N16-induced inhibition of VLP budding but had no detectable impact on the drug-induced Gag instability (Supplementary Fig. 11, Fig. 3, respectively). In contrast, the N16-induced Gag instability was blocked by treatment with MG132 (Supplementary Fig. 4). Interestingly, the proteasome inhibitor, bortezomib[33], a dipeptide which reversibly inhibits the chymotrypsin-like activity at the β5-subunit of the proteasome (PSMB5) had no apparent impact on the N16-induced inhibition of Gag budding and instability. MG132 is a structurally and functionally unrelated proteasome inhibitor which reversibly blocks all activities of the 26S proteasome but is not as specific as bortezomib[33]. MG132 is known to have off-target effects, e.g., it inhibits calpain and clasto-lactacystin β-lactone which inhibits cathepsin A. Although Schwartz et al.[56] showed that treatment of target cells with the proteasome inhibitors MG132 and lactacystin increased the early steps of HIV infection, Schubert et al.[19] demonstrated that virus assembly, maturation and budding require an active proteasome system. (For that reason, we did not conduct 'in-cellulo' MG132 treatments). Interestingly, like N16, pre-treatment of HIV-1 infected cells with MG132 enhances its inhibitory effect[19], suggesting that it, like N16, targets an early event in the budding pathway. However, in contrast to N16 treatment, MG132 treatment results in ultrastructural changes in budding virions similar to those resulting from mutations in the PTAP Late assembly domain[19], while the impact of N16 appears to be at an earlier stage. In any event, the observation that MG132 (NEM contribution to be defined) can oppose the N16-induced Gag instability while bortezomib did not, makes it unlikely that a specific complex like the proteasome is responsible. Consistent with this, we observed that the Gag lability could also be prevented by lysing cells in the presence of a mixture of protease inhibitors, as described in the text. At this point, it is not clear whether the Gag-destabilizing effect of N16 observed in 293T but not HeLa cells is an "off-target" effect of N16 in 293T cells or is in some way linked to an indirect impact on Tsg101 function in those cells.

We propose a Tsg101 chaperone function that is based on the participation of the Ub-binding pocket in temporal and/or spatial balancing of Gag ubiquitination and deubiquitination during budding. This adds Tsg101 to a list that includes proteins like BAG[57], Cdc48/97[58], and PDCL3[59] that fulfill their chaperoning function by influencing ubiquitination. Through this chaperone function, Tsg101 makes a previously unappreciated contribution to virus budding that appears to be required early in the budding process and distinct from the recruitment of ESCRT-III that is critical to HIV egress. Moreover, this function is required whether or not the PTAP pocket in the Tsg101 protein is engaged, as evident from P7L susceptibility. Possibly, as the binding is covalent, the F15/N16-modified Tsg101 might exert a *trans*-dominant-negative interfering influence on a function required for Alix-mediated P7L budding and thereby affect egress even if direct binding does not normally occur. That both Tsg101-driven and Alix-driven budding was inhibited by an agent (N16) that disrupts Tsg101 Ub-binding activity implies a requirement for this Tsg101 chaperone function in a fundamental aspect of budding.

Targeting of the Gag-Tsg101 interaction for inhibition of HIV budding has been an active field over the last 15 years, mostly focusing on interference with PTAP binding[60–63]. Theoretically, a synthetic peptide that mimics the PTAP motif could compete for the PTAP binding pocket of Tsg101 in cells. The wild-type synthetic peptide [5]PEPTAPPEE[13] displays a low binding affinity to Tsg101 in vitro, with a $K_d$ of 54 μM[64]; however, a significant increase in affinity can be achieved by introducing a bulkier and more hydrophobic group at the first proline in the PTAP motif[62, 64]. Although high-resolution structures for the highest affinity complexes have not yet been elucidated, docking studies indicate that these PTAP inhibitors bind to the hydrophobic surface of Tsg101 around residue T56[62]. Interestingly, this binding interface overlaps with the N16 interface, underscoring the importance of that hydrophobic surface. Cyclic peptides have also been shown to inhibit the Gag-Tsg101 interaction[60]. Several cyclic peptides structurally unrelated to PTAP but exhibiting high Tsg101 binding affinity inhibited HIV-1 VLP release >60% when covalently attached to the HIV Tat protein to facilitate their passage through the cell membrane. These peptides also inhibited the Tsg101-PSAP-Hrs interaction but had no apparent effect on EGFR degradation at low concentrations. Our studies demonstrate the ability of small molecules like F15 and N16 to interfere with a previously unrecognized Tsg101 contribution to budding. Our findings suggest that F15/N16 could most likely be used at higher concentrations than these PTAP-based inhibitors without adversely affecting normal cellular function. Indeed, the dose of N16 provided to humans is 40 mg. A concentration of 7 μg ml⁻¹ is achieved in human plasma[65], a level that is equivalent to 20 μM of the drug. The lowest effective concentration that we achieved in our tissue culture assays when N16 was provided under the optimal conditions described in the text (c.f. Fig. 1) was 25–50 μM. The current availability of the F15/N16 compounds in diverse long-acting slow-release formulations is another feature adaptable to further refined development of the F15/N16 compounds that could address issues pertaining to drug regimen adherence. Most important to the goal of targeting Tsg101 for development of next-generation antiviral agents, the structure of the N16-Tsg101 interaction is of sufficiently high resolution to be used in further development of improved Tsg101 inhibitors. Thus, findings described in this report present the Tsg101 UEV Ub binding activity as a target that can be exploited for antiviral drug design.

## Methods

**Plasmids and reagents**. pNL4-3ΔEnv, pIIIB Env3-1, pCMV-Gag-HA encoding HIV-1 Gag C-terminally tagged with hemagglutinin were previously described[31]. Plasmid encoding the HIV-1 Gag-Herpes Simplex virus deubiquitinating enzyme chimeric protein has been described[22]. pCMV-Gag-EGFP encoding HIV-1 Gag C-terminally tagged with green fluorescent protein (GFP) and pLLEXP1-hTsg101-myc encoding full-length human Tsg101 C-terminally tagged with myc were previously described[54]. pLLEXP1-hTsg101-myc and pLLEXP1-hTSG101-FLAG (siRNA resistant)[41] were used as template for site-directed mutagenesis to generate single amino acid substitution mutants: C73A and C87A (oligos used to introduce the mutation for C73A: 5′-CTTATAGAGGTAATACATACAATATTCCAA-TAGCCCTATGGCTACTGGACA-3′; for C87A: 5′CATACCCATA-TAATCCCCCTATCGCTTTTGTTAAGCCTACTAGTTCAA3′). Previously designed siRNA (targeting) directed against *Tsg101* nucleotides 410–434 (5′-aggacgagagaagactggaggttca3′) and siRNA (non-targeting) with the reverse sequence (5′-acttggaggtcagaagagagcagga3′)[41] were custom synthesized by Dharmacon. Primary antibodies were: Rb anti-CA[54] (1:1000); Rb anti-FLAG (1:1000, Sigma, F7425); Rb anti-EGFR (1:250, Santa Cruz Biotechnology, sc-03); mouse anti-HA (1:500, BioLegends, 901501); anti-actin (1:1000, Sigma, A4700); mouse anti-Tsg101 (1:500, GeneTex, GTX70255); mouse anti-myc (1:200, Santa Cruz Biotechnology, sc-40); secondary antibodies were: goat anti-mouse IgG Alexa Fluor 680 (1:2000, Molecular Probes, A21057); Texas Red tagged goat anti-mouse IgG (1:200, Invitrogen, T862); and goat anti-rabbit IRDye800 (1:2000, Rockland, 611-132-122). Chemicals: N-ethylmaleimide (NEM) and carbobenzoxyl-leucine-leucine-leucinal (MG132) (Sigma); bortezomib (Selleck Chemicals).

**Transfection and assays for particle budding and infectivity**. 293T (ATCC CRL-3216) and HeLa (ATCC CCL-2) cells were grown in Dulbecco's modified Eagle medium supplemented with fetal bovine serum (10%) and antibiotics (1%) to 70% confluency at 37 °C prior to drug treatment or transfection. Tissue culture

media was aspirated and replaced with control or treatment media prior to transfection unless stated otherwise in figure legends. Transfection was done using XtremeGene reagent (Roche) for DNA or Lipofectamine 2000 for siRNA (Invitrogen). For production of virus particles, cells were transfected with pNL4-3-ΔEnv and pIIIB Env3-1 plasmids and for production of virus-like particles (VLP), were transfected with a Gag-encoding construct specified in figure legends. After 24 h, tissue culture media was collected and passed through a 0.45 micron filter; cells were scraped with a rubber policeman, rinsed with PBS and pelleted. For specific infectivity, media-associated p24 was determined by ELISA (Immunodiagnostics, Inc.) and equivalent amounts of p24 were used to infect HeLa-CD4$^+$-LTR-βgal cells for infectivity measurement by the multinuclear activation of a galactosidase indicator (MAGI) assay. For VLP isolation, filtered media was centrifuged through a 20% sucrose cushion at 22,000×$g$ for 90 min at 5 °C and the pellet fraction saved for analysis. Cell pellets were lysed with either Triton X-100 buffer (50 mM Tris, pH 7.4, 137 mM NaCl, 1.5 mM MgCl$_2$, 1 mM EDTA, 1% Triton X-100) or RIPA buffer with complete mini protease inhibitor cocktail (Roche) and, where indicated in the text, additional inhibitors. VLP and cell lysate samples were analyzed by western blotting. Protein bands were visualized using an infrared-based imaging system (Odyssey, LI-COR Biotechnology) and band intensities measured using the Li-Cor Odyssey software version 2.1.15. Western blotting result panels were composed from uncropped blot images in Supplementary Fig. 12. Virus particle release efficiency was calculated [VLP signal intensity/(VLP signal intensity + cell lysate signal intensity)]. Quantification analyses plot the data mean with error bars signifying ±1 standard deviation (SD).

**IC$_{50}$ calculation.** Cells were grown in 96-well trays for 24 h with increasing levels of drug. Metabolic activity was measured using WST-1 colorimetric assay (Roche) where dye formed from a tetrazolium compound and an electron coupling reagent directly correlates to the number of metabolically active cells in the culture. IC$_{50}$ values were calculated (Prism 6, Graph Pad Software Inc.).

**Fluorescence microscopy.** HeLa cells grown on cover slips were transfected with pCMV-Gag-EGFP alone or together with pLLEXP1-hTsg101-myc. Cells were fixed in 4% formaldehyde (Sigma) and permeabilized in 0.1% Triton X-100. Tsg101 was detected in the samples by indirect immunofluorescence using anti-Myc Mab and Texas Red anti-mouse IgG. Nuclei were stained with Hoechst. Images were captured on an inverted fluorescence/differential-interference contrast (dic) Zeiss Axiovert 200 M deconvolving fluorescence microscope operated by AxioVision Version 4.5 (Zeiss) software and deconvolved by using the constrained iterative method (AxioVision). Protein co-localization was assessed in 40 or more cells by determination of Pearson's coefficient of correlation[39] using Image J software and regarded as significant when a value of 0.6 or higher (equivalent to a 95% level of confidence for that number of cells) was observed.

**Electron microscopy.** 293T cells grown on ACLAR film that have been transfected and treated as described were fixed in 2.5% EM grade glutaraldehyde in PBS, soaked in 2% osmium tetroxide, dehydrated in a graded series of ethyl alcohol solutions and embedded in Durpan resin. Thin sections of 80 nm were counter-stained with uranyl acetate and lead citrate and viewed with a FEI Tecanal BioTwinG2 electron microscope.

**Production of recombinant Tsg101 UEV domain.** N-terminally His$_6$-tagged Tsg101 UEV domain (residues 2–145) was encoded in a pET-28b vector (Novagen—EMD Millipore), which included a TEV protease cleavage site (His$_6$-TEV-Tsg101$^{2–145}$). Tsg101 protein was expressed in Rosetta 2 (DE3) pLysS *Escherichia coli* competent cells (EMD Millipore) grown in M9 medium containing 50 mg L$^{−1}$ kanamycin (Sigma) and 34 mg L$^{−1}$ chloramphenicol (Sigma), supplemented with $^{15}$NH$_4$Cl and either natural abundance glucose or [U-$^{13}$C]-glucose to obtain $^{15}$N-Tsg101 UEV and $^{15}$N/$^{13}$C-Tsg101 UEV, respectively. Cell pellets were solubilized in buffer containing 100 mM Tris, 300 mM NaCl, 20 mM imidazole, pH 7.5, and a complete EDTA-free protease inhibitor cocktail tablet (Roche). Lysozyme was added at a final concentration of 100 μg mL$^{−1}$ and the solution was stirred at room temperature for 1 h.

The resulting solution was passed twice through an emulsifier to break open the cells. Following centrifugation (40,000 r.p.m., 1 h, 20 °C), the supernatant was adjusted to 0.5 M NaCl, then passed through an immobilized nickel ion affinity chromatography column (HisTrap FF, GE Healthcare) equilibrated with 100 mM Tris, 0.5 M NaCl, pH 7.5, and the protein eluted using an imidazole gradient (20 mM−0.5 M imidazole). Tobacco etch virus (TEV) protease was then used to cleave the N-terminal His$_6$ tag from Tsg101, at a ratio of 1:100 w/w Tsg101:TEV, at room temperature, 16 h, while simultaneously dialyzing against buffer containing 100 mM Tris, 0.5 M NaCl, 20 mM imidazole, pH 7.5. A second nickel column (same conditions) separated TEV and the His$_6$ tag from Tsg101, followed by size-exclusion chromatography (HiLoad 16/60 Superdex 75 pg, GE Healthcare) for final purification. Cleavage by TEV protease resulted in a non-native glycine at the N-terminus (residue 1). NMR samples contained ~0.6 mM Tsg101 UEV, 20 mM potassium phosphate (pH 5.8), 50 mM NaCl, and 8% $^2$H$_2$O. For the $^{13}$C-aromatic, $^{13}$C-aliphatic, and $^{13}$C-filtered NOESY experiments, the sample was exchanged into the same buffer dissolved in 99.96% D$_2$O (Cambridge Isotope Laboratories).

**Production of recombinant wild-type ubiquitin.** Wild-type ubiquitin was encoded in a pET-21a vector (Novagen—EMD Millipore), expressed in BL21-Gold (DE3) *E. coli* competent cells (Agilent) grown in LB medium containing 100 mg L$^{−1}$ ampicillin (Sigma), and purified according to a protocol described by Lazar et al.[66]. Cell pellets were solubilized in buffer containing 50 mM Tris pH 7.5 and a cOmplete EDTA-free protease inhibitor cocktail tablet (Roche). The resulting solution was passed three times through an emulsifier to break open the cells and then incubated at room temperature with 10 μL of benzonase to digest DNA. Following centrifugation (40,000 r.p.m., 1 h, 20 °C), the supernatant was acidified using acetic acid to a final pH of 4.3, then stirred for 2 h at 4 °C. Following centrifugation (40,000 r.p.m., 1 h, 4 °C), the supernatant was dialyzed against buffer containing 50 mM sodium acetate, pH 4.5 (2 kDa MWCO). Following further centrifugation (40,000 r.p.m., 1 h, 4 °C), the supernatant was loaded onto a HiTrap S.P. H.P. (GE Healthcare) ion exchange column equilibrated with 50 mM sodium acetate pH 4.5 and eluted over a gradient from 0 to 1 M sodium chloride (eluted around 300 mM NaCl). Final purification was carried out using size exclusion gel filtration (S75 26/60) equilibrated with 50 mM sodium phosphate, pH 7.

**Preparation of Tsg101 UEV-N16 complex.** N16 (20 mM solution, DMSO) was added to Tsg101 UEV domain in a 10:1 ratio (N16 excess). Complete formation of the N16 UEV complex was observed at 2 h by following chemical shift perturbations, which coincided with a red coloring of the solution. Since the complex was covalent, excess unbound N16 and DMSO could be removed by Amicon Ultra centrifugal filtration (MWCO 10 kDa, Millipore).

**Liquid chromatography-mass spectrometry.** The intact mass and purity of the protein samples were confirmed by LC–MS (Agilent 6224 ESI-TOF LC-MS). LC–MS confirmed 99% $^{15}$N-labeling (16,727.8 Da for $^{15}$N-Tsg101 UEV vs expected 16,729.2 Da for 100% $^{15}$N-labeling) and 98% $^{13}$C-labeling (17,471.5 Da for $^{15}$N/$^{13}$C-Tsg101 UEV assuming 99% $^{15}$N-labeling vs expected 17,488.2 Da for 100% $^{15}$N- and $^{13}$C-labeling). Covalent attachment of N16 was also confirmed by LC–MS (17,057.2 Da for N16 $^{15}$N-Tsg101 UEV vs expected 17,057.2 Da, assuming loss of oxygen associated with rearrangement).

**NMR binding studies.** For binding studies using NMR spectroscopy, the following complexes were made: Tsg101 UEV-N16, UEV-PTAP, UEV Ub, UEV-N16-PTAP, and UEV-N16-Ub, all in Tsg101 NMR buffer: 20 mM potassium phosphate, 50 mM NaCl, pH 5.8, with UEV concentration of 200 μM. PTAP (Ace-NFLQSRPEPTAPPEE-CONH$_2$, Bio-Synthesis, Texas), is a synthetic peptide based on residues 15–16 of the HIV-1 Gag$^{WT}$ spacer peptide 2 ($^{15}$NF$^{16}$) and 1–13 of the HIV-1 Gag$^{WT}$ p6 sequence ($^1$LQSRPEPTAPPEE$^{13}$), which was dissolved in Tsg101 NMR buffer at a concentration of 1 mM before addition to Tsg101 UEV or UEV-N16 at a final 1:1 ratio. Ub (wild-type, unlabeled ubiquitin, 936 μM) was dialyzed against Tsg101 NMR buffer along with UEV or UEV-N16 prior to mixing at a final 1:1 ratio.

**Chemical shift perturbations.** The assignment of Ub in complex with Tsg101 UEV and UEV-N16 was carried out using chemical shift titrations, by measuring HSQC spectra at a 1:0, 1:0.5, and 1:1 ratio of UEV or UEV-N16 to Ub. The assignment of the UEV-PTAP complex was carried out using a $^{15}$N-edited NOESY spectrum and by comparison with the BMRB chemical shifts previously deposited for the structure of Tsg101 UEV in complex with a PTAP peptide[11] (BMRB: 5532, PDB: 1M4Q, 1M4P). The assignment of the UEV-N16-PTAP complex was assigned by direct comparison with the UEV-N16 and UEV-PTAP chemical shifts, for residues in the N16 and PTAP binding sites, respectively. All chemical shift perturbations were calculated according to the following equation: $\sqrt{[0.5((H_{complex}-H_{free})^2 + (\alpha(N_{complex}-N_{free}))^2)]}$, where $\alpha$ is a scaling factor[67] equal to 0.13, calculated from $\alpha = (H_{max}-H_{min})/(N_{max}-N_{min})$, where $H_{complex}$ and $N_{complex}$ describe the H and N chemical shifts in the complexed form, $H_{free}$ and $N_{free}$ describe the H and N chemical shifts in the free form, $H_{max}$ and $N_{max}$ describe the largest chemical shifts from the $^1$H/$^{15}$N-HSQC spectrum of Tsg101 in the free form, and $H_{min}$ and $N_{min}$ describe the smallest chemical shifts. Additionally, chemical shift perturbations involving two ligands, e.g., Tsg101-N16 with PTAP or Ub, were calculated in an analogous manner. The cutoff for large chemical shift perturbations was 1.5 standard deviations from zero.

**NMR spectroscopy.** NMR data were acquired at 300 K on Bruker 600, 800, and 900 MHz spectrometers, each equipped with a cryogenic probe, using $^{13}$C/$^{15}$N-Tsg101 in complex with natural abundance N16. Spectra were processed using NMRPipe[68] and analyzed using CCPN Analysis 2.4.1[69]. Assignments were completed using standard triple-resonance techniques, with data obtained from the following three-dimensional experiments: HNCACB and CBCA(CO)NH for backbone assignment[70, 71], C-dipsi-(CO)NH, H-dipsi-(CO)NH, and $^{15}$N-edited NOESY for side-chain assignment[70, 72], $^{15}$N-edited NOESY, $^{13}$C-edited NOESY (aliphatic and aromatic collected separately) for protein-protein NOE assignments[72], and $^{13}$C-filtered NOESY for protein-ligand NOE assignments, of which the latter exploited differences in isotopic labeling between the ligand (natural abundance, 99% $^{12}$C) and the protein (98% $^{13}$C)[73]. Two conformations of the protein-ligand complex were visible in NMR spectra, with the apparent

populations changing over time. The first conformation to appear (the kinetic product) disappeared over time until only the second conformation was visible (the thermodynamic product). Both conformations caused similar chemical shift perturbations and are likely to result from slight differences in ligand orientation within the same Tsg101 binding site. We chose to solve the structure of the more stable thermodynamic product since it would result in a higher quality structure.

**Tsg101 UEV-N16 structure calculations.** The structure of the Tsg101 UEV domain in complex with N16 was calculated using Xplor-NIH 2.44[74, 75]. Simulated annealing was used in combination with NOE distance restraints, dihedral angle restraints, and hydrogen bond restraints. Distance restraints were calculated from NOE peak heights using CCPN Analysis 2.4.1[69]. Dihedral angle restraints were derived from backbone chemical shift data using TALOS[N76]. Hydrogen bond restraints were obtained by predicting secondary structure propensity using MICS[77] and comparing with known hydrogen bonds from the Tsg101 UEV domain structure in the free form[12] (PDB: 1KPP). 200 structures were calculated using simulated annealing, of which the 20 with lowest energy were used for further refinement. During the secondary refinement stage, 100 structures were calculated, with the inclusion of the Xplor-NIH 'repel' energy term to avoid atomic clashes, and a 'refRMSD' term to prevent calculated structures from straying too far from the lowest energy structure derived using simulated annealing. The 20 structures from the final refinement that were lowest in energy were chosen for the structural bundle presented here. Procheck Ramachandran statistics were calculated for all residues using the Protein Structure Validation Suite web server[78]: most favored (91.9%), additionally allowed (7.3%), generously allowed (0.3%), disallowed (0.5%).

**Small molecule library for high-throughput screening.** A custom small molecule library was formed for high-throughput screening (HTS) using the commercially available MicroSource Spectrum collection plus additional natural products selected from the MicroSource catalog and other sources. Altogether >80,000 compounds were screened using the fluorescence thermal shift assay described below. The library included approved drugs, natural products, and compounds with established and diverse bioactivity. Seven compounds found to bind the Tsg101-UEV fragment in the assay were subsequently examined in cell toxicity, ELISA, and tissue culture budding assays.

**Fluorescence thermal shift screening.** The recombinant Tsg101 fragment (amino acids 2–145), prepared as described above (but without label) has a thermal unfolding profile suitable for using FTS as a primary screen assay in HTS (Supplementary Fig. 1). A fluorescence dye Sypro-Orange (Invitrogen) was used for assay detection. The dye is excited at 473 nm and has a fluorescence emission at 610 nm. The dye binds to hydrophobic regions of a protein that are normally buried in a native protein structure. When a protein is unfolded, the dye interacts with exposed hydrophobic regions and the fluorescence intensity increases significantly over that observed in aqueous solution. The Tsg101 fragment was pre-mixed at a concentration of 2 µM with a 5X concentration of Sypro-Orange in Hepes buffer (100 mM HEPES, 150 mM NaCl, pH 7.5). 10 µl of the protein-dye mix was added to an assay plate and 10–50 nl of compound, equal to 10–50 µM, were added with an acoustic transfer robot Echo550 (Labcyte, CA). The plate was shaken to ensure proper mixing and then sealed with an optical seal and centrifuged. The thermal scan was performed from 20 to 90 °C with a temperature ramp rate of 0.5 °C min$^{-1}$. The fluorescence was detected on a real-time PCR machine CFX384 (Bio-Rad Laboratories). Comparison of the thermal denaturation profile for Tsg101-UEV in the presence and absence of N16 (top panel) or F15 (bottom panel) revealed destabilization of the native protein structure, indicating that the compound interacted with Tsg101-UEV.

**Data availability.** Coordinates have been deposited in the Protein Data Bank with accession code 5VKG, while NMR chemical shifts and structural restraints have been deposited in the Biological Magnetic Resonance Data Bank with accession code 30285. Other data are available from the corresponding authors upon request.

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

## Acknowledgements

We thank Dr. Fadila Bouamr for providing the construct encoding the HIV-1 Gag-Herpes Simplex virus deubiquitinating enzyme chimeric protein; Sara F. Dunne of the Northwestern U High Throughput Screening (HTS) Facility for assistance with high-throughput screening; Susan van Horn (Stony Brook University Central Microscopy Core Facility) for electron microscopy services. We also thank Dr. Duck-Yeon Lee of the Biochemistry Core Facility and Yi He of the Laboratory of Structural Biophysics at the National Heart, Lung and Blood Institute (NHLBI) of the National Institutes of Health (NIH) for expertise and advice regarding LC–MS and for the growth of *E. coli* for production of isotopically-labeled Tsg101, respectively. We also acknowledge Dr. Guil-lermo A. Bermejo, Dr. Yong-Sok Lee, and Dr. Charles D. Schwieters at the Center for Information Technology at the NIH for help with preparing structure files for N16 and for help with structure calculations using Xplor-NIH. These investigations were supported by funds from the NIH National Institute of General Medical Sciences (NIGMS) R01 GM111028 and from the NHLBI (U01HL127522) and the New York State Department of Economic Development (C140151) and by the Center for Biotechnology, Stony Brook University to C.A.C.; NHLBI (U01HL127522) and NHLBI Intramural Research Programs to N.T.; Northwestern Memorial Hospital Dixon Innovation Grant and Chicago Biomedical Consortium/HTS Supplemental Grant to J.L. and C-H.L.; Lurie Cancer Center grant #P30CA060553 to C-H.L.; RCMI grant G12MD007602 and R21 NS105577 to M.D.P. The work is related to pending patent applications which name some of the authors as inventors.

## Author contributions

M.S. conducted NMR structural analysis, wrote the paper; L.S.E., conducted compound cell-based studies and biochemical analysis, coordinated infectivity studies, wrote the paper; S.W., conducted biochemical analysis; M.K., conducted infectivity assays; M.-P.S., developed recombinant Tsg101-UEV preparation protocol; C-H.L., conducted high-throughput compound screening; M.D.P., supervised infectivity studies; J.L., initiated high-throughput library screening that identified described compounds; N.T., supervised

structural analysis; C.A.C., supervised overall project development, coordinated structural, biochemical, cell-based, and infectivity studies, wrote the paper.

## Additional information

**Competing interests:** C.A.C. and J.L. are co-inventors on a patent application which includes subject matter from this manuscript. The remaining authors declare no competing financial interests.

