## [Peer Review File · Nature Communications]

Reviewers' comments:

Reviewer #1 (Remarks to the Author):

In this report, Strickland and colleagues report on the identification of a compound, N16, that directly binds to the ESCRT-I subunit Tsg101 and, in doing so, destabilizes HIV-1 Gag and disrupts particle assembly. Based on the Gag-destabilizing effect of this compound, the authors claim to have identified a novel Tsg101 chaperone function.

The authors provide some interesting results on the effects of a Tsg101-binding small molecule. These results are novel, and the mechanism of action of N16 is worthy of careful examination. The structural components of the work are nicely done. However, there are a number of limitations with the data and their interpretation that undermine the story. Specific comments are provided below.

1. If Tsg101 indeed serves a chaperone function such that its binding to Gag protects Gag from degradation, as the authors argue, then other approaches that block Gag-Tsg101 binding would also destabilize Gag in a manner analogous to that observed here with N16 treatment. The Gag-Tsg101 interaction has been disrupted in many studies in various ways (Tsg101 depletion, PTAP mutation, dominant-negative Tsg101 expression, etc) and this destabilization phenotype has not been observed. This issue, which is never raised by the authors, suggests that the the Gag-destabilizing effect of N16 may be an off-target effect of the compound.
2. The authors provide no information about the screen that led to N16. Also, the concentrations used here are very high (up to 50 and 100 μM). Because N16 is undergoing clinical trials for other purposes, the authors should comment, if possible, on whether the concentrations used here are physiologically attainable in vivo. I would suspect not.
3. The authors argue repeatedly that N16 treatment leads to Gag polyubiquitination, but they don't actually show this.
4. Based on the data in Fig. 1a, the authors argue that N16 induces the classical late domain defect (retention of budding particles at the plasma membrane). However, this is not convincing in several respects. First, they show EM images of only a couple particles. Such late-stage particles are commonly observed in virus-producing cells. This effect needs to be quantified. Second, the biochemical correlate of budding arrest – accumulation of p41 and p25 Gag processing intermediates – is not demonstrated.
5. Also in Fig. 1, the efficiency of VLP production cannot be accurately quantified without showing the corresponding cell lysates. If budding is being targeted by N16 then the release efficiency should be reduced.
6. Fig. S2. Panel a does show levels of Gag in the cell lysate. No difference is seen +/- N16. This is inconsistent with the Gag-destabilizing effect shown in other figures. How can this be explained?
7. If N16 destabilizes Gag via Tsg101 binding, then it should have no effect on Gag stability in the absence of Tsg101. This can be tested easily with RNAi-mediated Tsg101 knock-down, which is quite easy to achieve. This would be a direct test of whether the Gag-destabilizing effect (seen in some figures but not others) is Tsg101-related.
8. There are several problems with the confocal microscopy data (Fig. 2, 5). First, the wholesale redistribution of Tsg101 to the plasma membrane is an artifact of Gag overexpression, particularly codon-optimized Gag which is expressed at very high levels (see Welsch et al. & Krausslich Traffic 2006). Second, the large puncta observed with Tsg101-myc are aberrant class E compartments resulting from Tsg101 overexpression. The formation of class E compartments upon Tsg101 overexpression (or overexpression of several other ESCRT subunits) has been reported widely (though not mentioned or cited here). It's not clear what the relevance of these structures is in terms of localization. Third, the authors need to show the Gag and Tsg101 panels separately, then the merged panel. Fourth, the authors need to provide Pearson coefficient of correlation (R-values) for Gag-Tsg101 colocalization +/- N16 for all their data rather than % of cells exhibiting colocalization, which is a highly subjective measure. Fifth, the authors should show Tsg101 localization in the absence of Gag, +/- N16.

9. Line 184, the authors state that N16 did not interfere with Gag accumulation on the plasma membrane. Doesn't this contradict the Gag-destabilizing effect of N16 that the authors show?
10. Fig. 3b. The data are very noisy. If one compares the reduction in VLP production at the lowest and highest N16 input, it looks similar between Gag-Ub and Gag-Dub.
11. Fig. 3a. The authors show that MG132 reverses the Gag-destabilizing effect of N16. What happens to VLP production under these conditions? Is there still a budding block? This would address whether N16 does in fact target budding (see above) or simply destabilizes Gag.
12. Fig. 5. These data are not convincing. Non-merged images need to be shown (as mentioned above). The C87A + N16 panel is too dim to see anything.
13. If Tsg101 residue C73 is the target of N16, then knocking down endogenous Tsg101 and providing an siRNA resistant version of the C73A mutant should render HIV-1 assembly/release insensitive to N16. Such an experiment would greatly strengthen the story.
14. Error bars are provided for many of the results, but I see no mention of how they were determined. How many independent repeats were done, etc? This is a basic element of any scientific report.

Reviewer #2 (Remarks to the Author):

The manuscript provides evidence for a new function of Tsg101 in HIV-1 replication, i.e. chaperone protection to HIV-1 Gag independent of its interaction with the PTAP motif. The authors also identified compounds that bind to the N-terminal UEV domain of Tsg101 and they solved the NMR structure of the Tsg101 UEV:N16 complex. They demonstrate that UEV:N16 interaction inhibits ubiquitin binding by the UEV domain and that this inhibition dramatically affects the assembly of the HIV-1 Gag protein. The findings of these study may set the basis for the development of new antiviral strategies.

The experiments were well designed and performed, results were thoroughly discussed and interpreted (the reviewer concentrated on the NMR work). Overall, the manuscript is well written and technically sound, but a number of points need to be addressed before the manuscript can become acceptable for publication in Nature Communications

Major concerns

1) Binding studies of Tsg101 UEV with N16, PTAB, and Ub: Why did the authors just look at the 1:1 complexes instead of carrying out complete titrations? How did they unambiguously assign the signals without titrations? Moreover, the chemical shift perturbations for the UEV:Ub interaction appear very small and only very few residues are affected which, in turn, do not form a continuous binding surface. These data may be improved conducting a complete titration. Overall, the NMR spectra of these binding studies should be presented, at least as Supplementary Information, and the reviewer recommends to conduct complete titrations as these may provide more comprehensive data.

2) Supplementary Table 1: the violations of the distance constraint appear high ($4.0 \pm 2.2 \text{ \AA}$), especially as the max. distance constraint violation is 0.83 \AA
The max. dihedral angle violation is very high (23.67°). Please comment on these values.

Minor points

- 1) When describing how they calculated chemical shift perturbations, the authors should explain what the variables mean, e.g. Hmax; line 474
- 2) The general procedure of the structure calculation should be described (how were distance/dihedral restraints derived, which programmes were used, how was the general procedure); lines 479-495
- 3) Figure 4: The chronological order should be adapted, i.e. the order of the panels should be

adapted to the order in which they are mentioned in the text; ideally G and H should be converted to A and B as the authors state in line 283 that they solved the complex structure

Legend of Fig. 4: The authors should explain exactly how they calculated the significance level for the chemical shift perturbations (using the standard deviation), e.g. in Material and Methods; the authors should also describe that the horizontal line represents the significance level; line 272

4) Line 283: "structure of the complex" should be more clearly defined, e.g. "structure of the N16:Tsg101 UEV complex

5) Brackets have to be opened, Line 175

6) Brackets have to be closed, Line 209

7) Brackets have to be adapted, Line 277

8) Brackets have to be closed, Line 332

Reviewer #3 (Remarks to the Author):

In their manuscript Strickland et al. identified two related small molecular inhibitors, F15 and N16, that exhibit antiviral activity against HIV-1. Originally developed for treatment of heartburn, one of those two compounds already passed clinical trials and is available under the tradename Nexium (Esomeprazol). Beside the intriguing observation that Tsg101 may exhibit novel activity on Gag, the use of Nexium, or related drugs, as 2nd clinical indication for antiretroviral therapy is innovative and of great potential. Directed against a cellular target, such an approach would even pave the way to novel treatment options, particularly overcoming drug-resistant viruses. Both inhibitors, F15 and N16, covalently modify Tsg101 via cysteine 73, which was convincingly shown by biochemical and NMR approaches. The group of Carol Carter has a long standing record in investigating the function of Tsg101 in the late stage of HIV-1 replication. It has been well established that Tsg101, as part of ESCRT, mediates the final abscission step of the budding virus from the host membrane. Consistent with that, the Tsg101 targeting drugs N16/F15 interfere with HIV-1 particle release. Intriguingly, they also inhibit the L-domain mutant HIV-1 GagP7L, which is insensitive to Tsg101 mediated virus release. In a convincing manner the authors concluded that the compounds must somehow interfere with a novel function of Tsg101. In an attempt to elucidate this obvious conundrum, they observed that these drugs make Gag instable, increase its polyubiquitination, and make it a bona fide substrate for the 26S proteasome.

In summary, the authors identified two antiretroviral compounds that specifically and efficiently target the UEV-domain of Tsg101. This led to new insights into the interaction of HIV-1 Gag and Tsg101. Independently of the academic progress, Tsg101 may be added to the growing list of cellular targets in antiretroviral therapy. However, some additional experiments listed below might help to confirm the observations made by the authors about the mode of action of those Tsg101 specific inhibitors.

Specific comments:

1. [line 152-155] VLP release of the L-domain mutant GagP7L remains sensitive to N16. The mutant GagP7L per se has a severe budding defect that can be rescued by overexpression of Alix. The experiments should be repeated in a context where virus release of GagP7L is restored to the level of wildtype Gag. If N16 influences virus release in a way that occurs independently of L-domain functions of Tsg101, overexpression of Alix should not influence the inhibitory activity of N16.

2. [line 202-203] The cytotoxicity data should be included in a supplementary figure. Both, N16 and F15, are SH reactive agents. Originally, F15 was used as competitive inhibitor of the enzyme CYP2C19. Treatment with 0.1 mM of an SH reactive reagent bear the risk of off-target activities. Although convincingly shown that the inhibitors selectively and specifically interact with Tsg101, any unspecific and potentially toxic adverse effect should be thoroughly excluded. For the lead experiments at least some of the standard cytotoxicity assays should be conducted (MTT assay, proliferation assay, annexin V staining). Furthermore, the effect on the cell number during a two week treatment should be controlled (Fig. S1). Even if it proves true that both inhibitors are highly

selective to the UEV-domain of Tsg101, it goes without saying that a number of vital activities of Tsg101 are known. Therefore, it must be convincingly shown that N16/F15 do not affect other cellular functions at the IC50 values applied for antiretroviral activity .

3. [line 241-243] The authors convincingly showed that, after cell lysis, Gag from N16 treated cells is less stable compared to Gag from untreated cells. Furthermore, the proteasome inhibitor MG132 and the alkylating reagent NEM stabilized Gag ex cellulo in lysates derived from N16 treated cells. The drug itself is SH reactive. MG132 is not a specific proteasome inhibitor since it forms hemiacetale adducts with freely available OH-groups. NEM itself just alkylates unspecifically. Therefore, it has to be demonstrated that N16 mediated degradation of Gag can be stabilized by specific proteasome inhibitors, like Velcade or Lactacystin, in a physiologically more relevant context (e.g. in cellulo).

4. [line 254] The increased level of polyubiquitinated Gag in the presence of N16/F15 has been postulated but not demonstrated. This can be easily done by co-expression of HA-tagged ubiquitin and immunoprecipitation of Gag. Furthermore, inhibitors of deubiquitinating enzymes, like PR596, should increase the effect of N16 on Gag. On the contrary, overexpression of Lys 48 mutants of ubiquitin should reduce the effect of N16. Same should be true for using an E1 thermosensitive cell lines. There are a number of controls indicated to validate the interesting observation of Tsg101 controlled polyubiquitination of Gag.

5. [line 313-318] The covalent modification of Tsg101 by N16 has been convincingly characterized: The Tsg101 mutant C73A did not bind to N16, and N16 does not influence co-localization of this mutant with Gag. Why did the authors not show the loss of function of this mutant in terms of treatment with N16/F15? In the same line, upon overexpression of Tsg101 C73A, which should function as a trans dominant-negative mutant, N16 should lose its negative impact towards Gag release.

6. Another control for the specificity of the effect would be to add reagents that control the oxidative state in the extracellular and intracellular milieu. For instance, the addition of antioxidative reagents like glutathione should overrule the N16 effect.

7. Immune recognition of HIV positive cells are mainly controlled in vivo by Gag-derived MHC I epitopes. Beside the antiretroviral effect, treatment with N16/F15 should also deploy a very interesting side-effect: augmentation of the entry of Gag into the ubiquitin, proteasome and MHC-I pathway. Therefore, the TCD8 response to Gag should be improved, which might be of interest, not only for vaccine development but also for immune control of HIV+ cells during ART. Therefore, it would be a beautiful add-on to the importance of the general finding if the authors could show that N16/F15 increase the MHC I presentation of Gag derived epitopes.

8. It is well established that proteasome inhibitors exhibit an antiretroviral activity by causing accumulation of polyubiquitinated Gag proteins. With this in mind, the anti-Tsg101 drugs should have a synergistic effect with proteasome inhibitors in terms of virus budding and replication capacity. It would also be nice if the authors could control this aspect.

9. Several p6 mutants have been described to exhibit an increased Gag polyubiquitination independently of L-domains. One of them, the S40F, has been intensively characterized by the authors. Those mutants, in particular since they have almost normal virus release, should not, or at least less respond to N16/F15 treatment.

Some of these control experiments should be conducted in order to validate the very interesting finding described in this manuscript.

Minor points:

1. [line 39] At least once, the abbreviation UEV should be written-out.

2. [Fig.1a, right, top] The electron microscopy pictures only show small sections of budding virions. Why do the authors not show a larger section to demonstrate differences of mock and N16 treated cells or GagP7L? In general, almost any stage of virus morphogenesis can be seen even for wt virus.
3. [line 175-177] The authors state that Gag and Tsg101 co-localize in small puncta <10 nm as measured by confocal microscopy. Since the microscope used by the authors is bound to the Abbe limit, a resolution below 200 nm should not be feasible. The authors should either use STED/PALM super-resolution microscopy or correct their statement.
4. [line 184-187] The authors show that Tsg101 and GagP7L do not co-localize. They should comment in the discussion how Tsg101 can influence GagP7L without interaction.
5. [line 414] misspelling of the word Hoechst.

Responses to Reviewers' Comments for Strickland et al

Reviewer #1 (Remarks to the Author):

In this report, Strickland and colleagues report on the identification of a compound, N16, that directly binds to the ESCRT-I subunit Tsg101 and, in doing so, destabilizes HIV-1 Gag and disrupts particle assembly. Based on the Gag-destabilizing effect of this compound, the authors claim to have identified a novel Tsg101 chaperone function.

The authors provide some interesting results on the effects of a Tsg101-binding small molecule. These results are novel, and the mechanism of action of N16 is worthy of careful examination. The structural components of the work are nicely done. However, there are a number of limitations with the data and their interpretation that undermine the story. Specific comments are provided below.

COMMENT 1: If Tsg101 indeed serves a chaperone function such that its binding to Gag protects Gag from degradation, as the authors argue, then other approaches that block Gag-Tsg101 binding would also destabilize Gag in a manner analogous to that observed here with N16 treatment. The Gag-Tsg101 interaction has been disrupted in many studies in various ways (Tsg101 depletion, PTAP mutation, dominant-negative Tsg101 expression, etc) and this destabilization phenotype has not been observed. This issue, which is never raised by the authors, suggests that the Gag-destabilizing effect of N16 may be an off-target effect of the compound.

RESPONSE:

(a) *Regarding the unexpected effect of N16 on budding:* The outcome of disrupting the Gag-Tsg101 interaction in various ways (Tsg101 depletion¹, PTAP mutation^{23,46,47}, dominant-negative Tsg101 expression^{30,47,48}) and the known inhibitory effects of these treatments (*i.e.*, tethered particles, defective maturational p25 to p24 processing, polyubiquitination and single released particles arrested in the immature state are indeed clearly distinct from those resulting from exposure to F15 and N16 (aggravated accumulation of 'Early' buds compared to the mock-treated sample; aggravated accumulation of 'Early' buds compared to the outcome of PTAP mutation; sensitivity maintained despite Alix rescue of PTAP mutation (*please see below*, Reviewer-3, comment-1). Additionally, N16 treatment did not increase the proportion of tethered particles nor prevent the maturation of released particles, as does PTAP mutation (*c.f.*, Figure 1). We also eliminated cell toxicity and demonstrated functional specificity. Thus, although unexpected, we conclude that the distinct outcomes observed following Tsg101-PTAP *versus* Tsg101-Ub disruption are bonafide reflections of interference with different contributions Tsg101 makes to the budding process. We believe that the Tsg101-Ub interaction provides a chaperone function but precisely how is not clear at this time. We have removed from the text the suggestion that Tsg101 binding to Gag protects Gag from degradation.

(b) *Regarding the Gag-destabilizing effect of N16,* we agree with the Reviewer: It may be an "off-target" effect of the compound manifested in 293T cells. In the original submission, we demonstrated that the addition of MG132/NEM to cells for just 1 hour prior to disrupting them

for lysate preparation was protective (longer treatment is inhibitory, as previously demonstrated, Schubert et al 2000). Thus, Gag loss occurred “ex-cellulo”, *i.e.*, Gag degradation occurred upon 293T cell lysis, after the supernatant containing the VLP was removed. In the revised manuscript, (i) we describe that the loss can be prevented by addition of a mixture of proteases during lysate preparation. (ii) We show that depletion of endogenous Tsg101 and replacement with a Tsg101 mutant that is resistant to N16 inhibition (Cys73Ala) prevented N16-induced inhibition of budding but not the loss of intracellular Gag (*Supplementary Figure S10*). (iii) We show that the Gag-destabilizing effect of N16 observed in 293T cells was not observed in HeLa cells (*Supplementary Figure S3a*). In contrast, the inhibitory effect of N16 on budding was observed in 293T (*Figure 1*), Jurkat (*Supplementary Figure S2*), and HeLa (*Supplementary Figure S3a*) cells. These issues are now discussed in the revised manuscript.

COMMENT 2a: The authors provide no information about the screen that led to N16.

RESPONSE: In the original submission, information about the screen leading to N16 identification was provided as supplemental information on lines 678 – 705. However, this was not noted in the main text and was therefore apparently not noticed by the Reviewer. We apologize for this oversight. In this revised manuscript, we have noted in the main text that a description of the screen is in the *Supplementary Information* section and added additional experimental data as *Supplementary Figure 1*.

In revised manuscript, lines 103-105: “F15 (esomeprazole) and N16 (tenatoprazole) are two related compounds identified in a screen of small molecules capable of binding to the UEV domain of Tsg101 (*Supplementary Figure S1*).”

COMMENT 2b: Also, the concentrations used here are very high (up to 50 and 100 μ M). Because N16 is undergoing clinical trials for other purposes, the authors should comment, if possible, on whether the concentrations used here are physiologically attainable in vivo. I would suspect not.

RESPONSE: The dose of N16 and its analogs provided for use and in trial is 40 milligrams. Seven microgram/ml is achieved in human plasma. This level is equivalent to a 20 micromolar concentration of the drug. The lowest effective concentration that we achieved in our tissue culture assays when N16 was provided under the optimal conditions described in the text (*c.f. Figure 1a*) was 25-50 micromolar---that is not so bad for a lead compound in this case! As stated in the manuscript, we consider these compounds as leads that can provide platforms for development of derivatives with increased permeability or tighter binding to Tsg101.

In revised manuscript, lines 483-486: “Most important to the goal of targeting this newly appreciated contribution of Tsg101 for development of next-generation antiviral agents, the structure of the N16-Tsg101 interaction is of sufficiently high resolution to be used in further development of improved Tsg101 inhibitors.”

COMMENT 3: The authors argue repeatedly that N16 treatment leads to Gag polyubiquitination, but they don’t actually show this.

RESPONSE: This was a suggested possibility. In the revised manuscript, we omit this suggestion.

COMMENT 4a: Based on the data in Fig. 1a, the authors argue that N16 induces the classical late domain defect (retention of budding particles at the plasma membrane). However, this is not convincing in several respects. First, they show EM images of only a couple particles. Such late-stage particles are commonly observed in virus-producing cells. This effect needs to be quantified.

RESPONSE: The reviewer is correct and we thank him/her for suggesting a more extensive analysis. In the revised manuscript, the effect of N16 on EM particle morphology was re-determined under optimal treatment conditions (*i.e.*, drug addition prior to DNA transfection) and quantitation is provided in *Figure 1c*.

COMMENT 4b: Second, the biochemical correlate of budding arrest – accumulation of p41 and p25 Gag processing intermediates – is not demonstrated

RESPONSE: Accumulation of p41 and p25 Gag processing intermediates is a correlate of budding arrest when the Gag-PTAP-Tsg101 interaction is disrupted (*documented in several published papers including, Garrus, J. E. et al. Cell 107, 55-65 (2001); Martin-Serrano, J. et al. Nat Med 7, 1313-1319 (2001); Hahn, S. et al. J Immunol 186, 5706-5718 (2011); Demirov, D. G. et al. Proc Natl Acad Sci U S A 99, 955-960 (2002) and others*). Our studies demonstrate that N16 does not disrupt PTAP binding (*c.f. Figure 4 and Supplementary Figures S5, S6 and S7*).

COMMENT 5: Also in Fig. 1, the efficiency of VLP production cannot be accurately quantified without showing the corresponding cell lysates. If budding is being targeted by N16 then the release efficiency should be reduced.

RESPONSE: As shown in Figure 1, positioning Gag for degradation upon cell lysis appears to be an aspect of N16 inhibition. This outcome is investigated in Supplementary *Figure S3* in the revised manuscript and conditions permitting determination of VLP release efficiency are shown. The release efficiency was reduced, consistent with the conclusion that N16 targets budding. This conclusion is also supported by analysis of particle budding using the electron microscope.

COMMENT 6: Fig. S2. Panel a does show levels of Gag in the cell lysate. No difference is seen +/- N16. This is inconsistent with the Gag-destabilizing effect shown in other figures. How can this be explained?

RESPONSE: The result is not “inconsistent”, just a part of the inhibition response range: We observed that cytoplasmic Gag accumulation was reduced to 20-54% of the control level in 15/17 independent trials. In the remaining two experiments, accumulation was comparable to control. This range has been noted in the text.

COMMENT 7: If N16 destabilizes Gag via Tsg101 binding, then it should have no effect on Gag stability in the absence of Tsg101. This can be tested easily with RNAi-mediated Tsg101 knock-down, which is quite easy to achieve. This would be a direct test of whether the Gag-destabilizing effect (seen in some figures but not others) is Tsg101-related.

RESPONSE: In a depletion-replacement experiment (*Supplementary Figure S10*), replacement of Tsg101 in the cell with a Tsg101 mutant where the residue (C73) that is targeted by the drug has

been mutated (C73A) rendered budding resistant to N16. Under this condition, Gag instability was still observed. As indicated in the response to Comment 1, we now conclude that the effect on the intracellular Gag reflects an “off-target” effect of N16 manifested in 293T cells.

COMMENT 8a: There are several problems with the confocal microscopy data (Fig. 2, 5). First, the wholesale redistribution of Tsg101 to the plasma membrane is an artifact of Gag overexpression, particularly codon-optimized Gag which is expressed at very high levels (see Welsch et al. & Krausslich Traffic 2006).

RESPONSE: We agree with the reviewer and Welsch, S. *et al. Traffic* **7**, 1551-1566 (2006) that there is no wholesale re-distribution of Tsg101 to the plasma membrane in HIV-1 infected cells. All these are suggesting that the amount of ESCRT proteins brought to the budding site represents a very small fraction of the total ESCRT population yet suffices for HIV release. We thank the Reviewer for highlighting this point.

COMMENT 8b: Second, the large puncta observed with Tsg101-myc are aberrant class E compartments resulting from Tsg101 overexpression. The formation of class E compartments upon Tsg101 overexpression (or overexpression of several other ESCRT subunits) has been reported widely (though not mentioned or cited here). It’s not clear what the relevance of these structures is in terms of localization.

RESPONSE: The text has been edited to acknowledge the large puncta as aberrant Class E compartments and to cite the appropriate references describing their formation following Tsg101 overexpression.

In revised manuscript, lines 189-192: “Previously shown aberrant enlarged (>200 nm in diameter) endosomal compartments induced by adventitious expression of the Tsg101 protein (or several other ESCRT subunits)^{30,31} were seen by fluorescence microscopy of cells expressing Tsg101 tagged with Myc (FIGURE 2, *panel a, top*).”

COMMENT 8c: Third, the authors need to show the Gag and Tsg101 panels separately, then the merged panel.

RESPONSE: *Figures 2 and 5* have been edited to show Gag, Tsg101, and merged panels separately.

COMMENT 8d: Fourth, the authors need to provide Pearson coefficient of correlation (R-values) for Gag-Tsg101 colocalization +/- N16 for all their data rather than % of cells exhibiting colocalization, which is a highly subjective measure.

RESPONSE: The figures have been edited to provide Pearson coefficient of correlation values for Gag-Tsg101 colocalization +/- N16 for each data set shown rather than % of cells exhibiting co-localization. We thank the Reviewer for suggesting this improvement.

COMMENT 8e: Fifth, the authors should show Tsg101 localization in the absence of Gag, +/- N16.

RESPONSE: Yes, of course: We apologize for this omission. *Figure 2* has been edited to show Tsg101-myc localization in the presence of Gag, +/- N16. The revised manuscript also includes a new *Supplementary Figure S4* that shows localization of endogenous Tsg101 +/- N16.

COMMENT 9: Line 184, the authors state that N16 did not interfere with Gag accumulation on the plasma membrane. Doesn't this contradict the Gag-destabilizing effect of N16 that the authors show?

RESPONSE: We thank the reviewer for this astute observation prompting a number of additional experiments (see *Supplementary Figure S3*) and gave us better understanding of the issue. We now know that Gag became susceptible to degradation only upon cell lysis.

COMMENT 10: Fig. 3b. The data are very noisy. If one compares the reduction in VLP production at the lowest and highest N16 input, it looks similar between Gag-Ub and Gag-Dub.

RESPONSE: We agree with the Reviewer that a comparison of Gag-Ub and Gag-DUb VLP production did not clearly illustrate the point we wished to make. In the revised manuscript, the text and *Figure 3* have been revised to make more clearly the point that N16 is targeting Gag's reliance on the Ub-binding function of Tsg101: (i) the text now explicitly presents the background & rationale for this hypothesis; (ii) a new *panel 3a* shows the deleterious effect of DUb on Gag VLP production, confirming previous studies; (iii) rather than comparing the effect of the drug on Gag-Ub and Gag-DUb, the figure now focuses on the Gag-DUb outcome in the presence and absence of N16. Quantitative analysis now compares the relative WT Gag and Gag-DUb responses, including VLP release efficiency. Triplicate samples are provided to demonstrate reproducibility.

COMMENT 11: Fig. 3a. The authors show that MG132 reverses the Gag-destabilizing effect of N16. What happens to VLP production under these conditions? Is there still a budding block? This would address whether N16 does in fact target budding (see above) or simply destabilizes Gag.

RESPONSE: Gag degradation occurs upon cell lysis. We found that the addition of MG132 to cells for just 1 hour prior to disrupting them for lysate preparation was protective (longer is inhibitory). Previous studies demonstrated that MG132 inhibits HIV-1 budding upon '*in cellulo*' treatment (Schubert et al 2000), therefore, we used another proteasome inhibitor to examine the Reviewer's question. Under the conditions of Bortezomib (Velcade) treatment, VLP production was inhibited by N16 and cell-associated Gag was vulnerable. We concluded that the Gag instability and the block to budding are both outcomes of N16 treatment in 293T cells but it is unlikely that a specific complex like the proteasome is responsible for the Gag degradation. It also seems unlikely that the two effects are directly linked since depletion of endogenous Tsg101 and replacement with a Tsg101 mutant that is resistant to N16 inhibition (Cys73Ala) prevented the N16-induced inhibition of budding but not the loss of intracellular Gag (*Supplementary Figure S10*). Only the block to budding was observed following N16 treatment of HeLa cells. (*Supplementary Figure S3*).

COMMENT 12: Fig. 5. These data are not convincing. Non-merged images need to be shown (as mentioned above). The C87A + N16 panel is too dim to see anything.

RESPONSE: *Figure 5* has been edited to show non-merged images with brighter signal intensities.

COMMENT 13: If Tsg101 residue C73 is the target of N16, then knocking down endogenous Tsg101 and providing a siRNA resistant version of the C73A mutant should render HIV-1 assembly/release insensitive to N16. Such an experiment would greatly strengthen the story.

RESPONSE: The prediction that providing a siRNA resistant version of the C73A mutant should render HIV-1 assembly/release insensitive to N16 and validation that it does indeed do this has been added to the revised manuscript as *Supplementary Figure S10*. MAGI assay was used to measure viral particles in the tissue culture media rather than typical Western analysis to provide greater confidence in quantifying the outcome. The figure shows the mean values obtained in two independent experiments.

COMMENT 14: Error bars are provided for many of the results, but I see no mention of how they were determined. How many independent repeats were done, etc? This is a basic element of any scientific report.

RESPONSE: Quantification analyses plot the data mean with error bars signifying plus or minus 1 SD. (Cumming, Fidler, and Vaux, 2007). This information has been added to the Materials & Methods section. Additionally, the number of independent repeats performed has been added to figure legends where appropriate.

Reviewer #2 (Remarks to the Author):

The manuscript provides evidence for a new function of Tsg101 in HIV-1 replication, i.e. chaperone protection to HIV-1 Gag independent of its interaction with the PTAP motif. The authors also identified compounds that bind to the N-terminal UEV domain of Tsg101 and they solved the NMR structure of the Tsg101 UEV:N16 complex. They demonstrate that UEV:N16 interaction inhibits ubiquitin binding by the UEV domain and that this inhibition dramatically affects the assembly of the HIV-1 Gag protein. The findings of these study may set the basis for the development of new antiviral strategies.

The experiments were well designed and performed, results were thoroughly discussed and interpreted (the reviewer concentrated on the NMR work). Overall, the manuscript is well written and technically sound, but a number of points need to be addressed before the manuscript can become acceptable for publication in Nature Communications

Major Points

COMMENT 1a: Binding studies of Tsg101 UEV with N16, PTAB, and Ub: Why did the authors just look at the 1:1 complexes instead of carrying out complete titrations? How did they unambiguously assign the signals without titrations?

RESPONSE: The binding of Tsg101 UEV with N16, PTAP, and Ub occurs on slow, slow/intermediate, and fast timescales, respectively, in relation to the timescale of the NMR experiment. For this reason, only the UEV-Ub and UEV-N16-Ub interactions could be assigned using titrations. As such, we decided to display the chemical shift perturbations at a 1:1 ratio to standardize the measurements between each of the data sets. The PTAP-UEV interaction was instead assigned using BMRB chemical shifts (ID: 5532) for the Tsg101 UEV domain in complex with a PTAP peptide (Pornillos et al., 2002, *EMBO J.* 21:2397-2406). We decided to supplement these chemical shifts with a ¹⁵N-NOESY to confirm and improve our assignments. UEV in the free form and the UEV-N16 complex were both assigned using HNCACB and CBCA(CO)NH spectra using standard triple-resonance assignment methods. The UEV-N16-PTAP complex was assigned by comparison with the UEV-N16 and UEV-PTAP assignments. We have updated the Methods section to clarify this. We have also taken the opportunity to improve the chemical shift perturbation figures (now Figure 4d, f, and h) by repeating the titrations at a higher concentration of Tsg101 (200 μM – previously 100 μM) and by changing from a histogram view to a connected scatter plot, which we feel shows more clearly the difference (and similarity) between CSPs measured in the presence and absence of N16. Additionally, in repeating the titrations, we found an error in the equation we used for the calculation of UEV-Ub and UEV-N16-Ub CSPs, which we have now corrected. Other data sets were not affected.

$$\Delta\delta = \sqrt{\frac{1}{2}[\delta_H^2 + \alpha(\delta_N)^2]} \quad (\text{original})$$

$$\Delta\delta = \sqrt{\frac{1}{2}[\delta_H^2 + (\alpha \cdot \delta_N)^2]} \quad (\text{corrected})$$

COMMENT 1b: Moreover, the chemical shift perturbations for the UEV:Ub interaction appear very small and only very few residues are affected which, in turn, do not form a continuous binding surface. These data may be improved conducting a complete titration.

RESPONSE: The chemical shift perturbations for the UEV-Ub interaction are indeed very small, owing to the low binding affinity (~500 μM, Pornillos et al., 2002, *EMBO J.* 21:2397-2406). The residues with large CSPs do not form a contiguous binding surface, which is expected, since the X-ray crystal structure of the Tsg101 UEV-Ub complex (PDB: 1S1Q, Sundquist et al., 2004, *Molec. Cell* 13:783-789) shows that Tsg101 binds to Ub with three loops, rather than a flat interaction surface. In the crystal structure, contacts are made between Ub and Tsg101 residues 41-47 (β₁-β₂ loop), 88 (β₄ strand), 91-93, 95-97 (β₄-β₅ loop), and 105-106 (β₅-α₃ loop), comprising 16 residues of the binding site in total (see Figure panel a, below). Using a cutoff of 1.5 SD above zero, we could observe large CSPs in each of three loops mentioned above (Figure panel b,

below). Those residues included 39-40, 43, 45-57 (β_1 - β_2 loop), 87 (β_4 strand), 89, 92, 98 (β_4 - β_5 loop), and 106-107 (β_5 - α_3 loop). As such, we decided that the cutoff, as well as the choice of protein:ligand ratio (1:1), was sufficient to describe the binding site well and to observe the changes caused by pre-incubation with N16.

Figure caption: Comparison of the ubiquitin-Tsg101 interaction as described by X-ray crystallography and NMR chemical shift perturbations. The X-ray crystal structure of Tsg101 (magenta) in complex with ubiquitin (yellow), showing Ub-binding Tsg101 residues in sticks and color, as determined by **A**) X-ray crystallography side-chain contacts (Sundquist et al., 2004, *Molec. Cell*, 13:783-789) and **B**) large chemical shift perturbations of Tsg101 and ubiquitin at a 1:1 ratio (cut-off 1.5 SD above zero). In both cases, PDB structure 1S1Q is used, and residues are color-coded as follows: β_1 - β_2 loop in green, β_4 strand and β_4 - β_5 loop in orange, and β_5 - α_3 loop in blue.

COMMENT 1c: Overall, the NMR spectra of these binding studies should be presented, at least as Supplementary Information, and the reviewer recommends to conduct complete titrations as these may provide more comprehensive data.

RESPONSE: We have provided HSQC spectra in Supplementary Information for all binding studies in Figure 4.

COMMENT 2a: Supplementary Table 1: the violations of the distance constraint appear high (4.0 +/- 2.2 Å), especially as the max distance constraint violation is 0.83 Å

RESPONSE: We thank the reviewer for noticing this error. We had intended to show the number of violations per structure (4.0 +/- 2.2), which should not have been described using Ångstroms. We in fact have no consistent violations above 0.5 Å and so we realized that this style of

violation reporting is not the clearest method. As such, we have decided to give an r.m.s.d. value instead, to match the other violations in Supplementary Table 1.

COMMENT 2b: The max. dihedral angle violation is very high (23.67 °). Please comment on these values.

RESPONSE: We agree that this is high. We took the opportunity to add a further refinement step to the UEV-N16 structure calculation, where we added an Xplor-NIH 'repel' energy term. 'repel' is a non-bonded repulsive term that scales the size of the atomic radii such that fewer clashes will occur (and true global minimum is more likely to be reached). Following this refinement step, we found that the large dihedral violations in the flexible C-terminus (residues 137-139) were no longer present, resulting in a maximum dihedral violation of 8.57°. We also removed two NOE peaks as noise, and reassigned two others. In addition to improved overall structural statistics (Supplementary Table 1), the MolProbity Clashscore is reduced from 40.06 to 16.38. We have resubmitted the structure to the PDB and BMRB, for which we have new codes (5VKG and 30285, respectively), as updated in the text. We have also updated the Methods section to include the new methodology. Figures 4a and 4b have been updated with the new structure and we have also taken the opportunity to change the color scheme so that it is easier to distinguish (red/blue rather than red/green).

Minor points

COMMENT 1: When describing how they calculated chemical shift perturbations, the authors should explain what the variables mean, e.g. Hmax; line 474

RESPONSE: We have added the description of these variables to the Methods section.

COMMENT 2: The general procedure of the structure calculation should be described (how were distance/dihedral restraints derived, which programmes were used, how was the general procedure); lines 479-495

RESPONSE: We apologize for this omission and have updated the Methods section to include this.

COMMENT 3a: Figure 4: The chronological order should be adapted, i.e. the order of the panels should be adapted to the order in which they are mentioned in the text; ideally G and H should be converted to A and B as the authors state in line 283 that they solved the complex structure

RESPONSE: We have updated the figure and caption as suggested.

COMMENT 3b: Legend of Fig. 4: The authors should explain exactly how they calculated the significance level for the chemical shift perturbations (using the standard deviation), e.g. in Material and Methods; the authors should also describe that the horizontal line represents the significance level; line 272

RESPONSE: We have updated the figure and Methods section as requested.

COMMENTS 4-8:

- 4) Line 283: “structure of the complex” should be more clearly defined, e.g. “structure of the N16:Tsg101 UEV complex
- 5) Brackets have to be opened, Line 175
- 6) Brackets have to be closed, Line 209
- 7) Brackets have to be adapted, Line 277
- 8) Brackets have to be closed, Line 332

RESPONSE: We have updated the text accordingly.

Reviewer #3 (Remarks to the Author):

In their manuscript Strickland et al. identified two related small molecular inhibitors, F15 and N16, that exhibit antiviral activity against HIV-1. Originally developed for treatment of heartburn, one of those two compounds already passed clinical trials and is available under the trade name Nexium (Esomeprazol). Beside the intriguing observation that Tsg101 may exhibit novel activity on Gag, the use of Nexium, or related drugs, as 2nd clinical indication for antiretroviral therapy is innovative and of great potential. Directed against a cellular target, such an approach would even pave the way to novel treatment options, particularly overcoming drug-resistant viruses. Both inhibitors, F15 and N16, covalently modify Tsg101 via cysteine 73, which was convincingly shown by biochemical and NMR approaches. The group of Carol Carter has a long standing record in investigating the function of Tsg101 in the late stage of HIV-1 replication. It has been well established that Tsg101, as part of ESCRT, mediates the final abscission step of the budding virus from the host membrane. Consistent with that, the Tsg101 targeting drugs N16/F15 interfere with HIV-1 particle release. Intriguingly, they also inhibit the L-domain mutant HIV-1 GagP7L, which is insensitive to Tsg101 mediated virus release. In a convincing manner the authors concluded that the compounds must somehow interfere with a novel function of Tsg101. In an attempt to elucidate this obvious conundrum, they observed that these drugs make Gag unstable, increase its polyubiquitination, and make it a bona fide substrate for the 26S proteasome. In summary, the authors identified two antiretroviral compounds that specifically and efficiently target the UEV-domain of Tsg101. This led to new insights into the interaction of HIV-1 Gag and Tsg101. Independently of the academic progress, Tsg101 may be added to the growing list of cellular targets in antiretroviral therapy. However, some additional experiments listed below might help to confirm the observations made by the authors about the mode of action of those Tsg101 specific inhibitors.

Specific comments:

COMMENT 1: [line 152-155] VLP release of the L-domain mutant GagP7L remains sensitive to N16. The mutant GagP7L per se has a severe budding defect that can be rescued by overexpression of Alix. The experiments should be repeated in a context where virus release of GagP7L is restored to the level of wildtype Gag. If N16 influences virus release in a way that occurs independently of L-domain functions of Tsg101, overexpression of Alix should not influence the inhibitory activity of N16

RESPONSE: We conducted the experiment ($n = 2$). Consistent with characterization of Alix rescue of PTAP mutant budding we found that (α) Alix increased P7L's VLP production (3-fold

and 2.5-fold, respectively: under the conditions of the experiment, these values reflect 100% restoration to WT level) and (b) Alix increased the p24/25 ratio. Nevertheless, Alix expression did not influence the inhibitory activity of N16. So, yes, the inhibitory effect of N16 does not require that Gag have an intact PTAP L domain. We think that we have made that clear in several places the revised manuscript. The more extensive analysis of budding morphology shown in the revised Fig 1 highlights the common feature of arrest at an early stage of budding in both N16 –treated WT and P7L NL4-3.

COMMENT 2a: [line 202-203] The cytotoxicity data should be included in a supplementary figure. Both, N16 and F15, are SH reactive agents. Originally, F15 was used as competitive inhibitor of the enzyme CYP2C19. Treatment with 0.1 mM of an SH reactive reagent bear the risk of off-target activities. Although convincingly shown that the inhibitors selectively and specifically interact with Tsg101, any unspecific and potentially toxic adverse effect should be thoroughly excluded. For the lead experiments at least some of the standard cytotoxicity assays should be conducted (MTT assay, proliferation assay, annexin V staining).

RESPONSE: Cytotoxicity data are now included in *Supplementary Figure S5*.

COMMENT 2b: Furthermore, the effect on the cell number during a two week treatment should be controlled (Fig. S1). Even if it proves true that both inhibitors are highly selective to the UEV-domain of Tsg101, it goes without saying that a number of vital activities of Tsg101 are known. Therefore, it must be convincingly shown that N16/F15 do not affect other cellular functions at the IC50 values applied for antiretroviral activity.

RESPONSE: The cell number was monitored by Trypan Blue assay during the two week treatment described in (what is now) *Supplementary Figure S2*. Not only was there no indication of high cell toxicity but we also demonstrate in the figure that the cells were capable of producing virus at high levels following cessation of drug treatment at the end of the two week period. *Supplementary Figure S5* (formerly Fig S2) provides evidence that N16 did not affect other cellular functions at the same IC50 values applied for antiretroviral activity.

COMMENT 3: [line 241-243] The authors convincingly showed that, after cell lysis, Gag from N16 treated cells is less stable compared to Gag from untreated cells. Furthermore, the proteasome inhibitor MG132 and the alkylating reagent NEM stabilized Gag *ex cellulo* in lysates derived from N16 treated cells. The drug itself is SH reactive. MG132 is not a specific proteasome inhibitor since it forms hemiacetale adducts with freely available OH-groups. NEM itself just alkylates unspecifically. Therefore, it has to be demonstrated that N16 mediated degradation of Gag can be stabilized by specific proteasome inhibitors, like Velcade or Lactacystin, in a physiologically more relevant context (*e.g., in cellulo*).

RESPONSE: *Supplementary Figure S3* in the revised manuscript shows the outcome when cells were exposed '*in-cellulo*' to Velcade (bortezomib). Treating cells throughout the period of N16 exposure failed to stabilize the cytoplasmic Gag. N16-mediated inhibition of VLP production

was not suppressed. We have removed from the text any suggestion that the proteasome is specifically involved in the N16 effect.

COMMENT 4a: [line 254] The increased level of polyubiquitinated Gag in the presence of N16/F15 has been postulated but not demonstrated. This can be easily done by co-expression of HA-tagged ubiquitin and immunoprecipitation of Gag. Furthermore, inhibitors of deubiquitinating enzymes, like PR596, should increase the effect of N16 on Gag. On the contrary, overexpression of Lys 48 mutants of ubiquitin should reduce the effect of N16.

RESPONSE: As noted above, we have removed from the text any suggestion that the proteasome is specifically involved in the N16 effect.

COMMENT 4b: Same should be true for using an E1 thermosensitive cell lines. There are a number of controls indicated to validate the interesting observation of Tsg101 controlled polyubiquitination of Gag.

RESPONSE: Using an E1 thermosensitive cell lines to test if the N16 effect is reduced sounds interesting, however, the cell lines we could identify are mouse fibroblast and expression of our constructs in rodent cell lines is very poor. In addition to this restriction, we have removed from the text any suggestion that the proteasome is specifically involved in the N16 effect, as noted above.

COMMENT 5a: [line 313-318] The covalent modification of Tsg101 by N16 has been convincingly characterized: The Tsg101 mutant C73A did not bind to N16, and N16 does not influence co-localization of this mutant with Gag. Why did the authors not show the loss of function of this mutant in terms of treatment with N16/F15?

RESPONSE: Lines 349-365 describe results shown in Figure 5, specifically panel b. Panel c in the Figure showed results of N16 effect on plasma membrane co-localization of Gag-C73A Tsg101 as a test of loss of function of the mutant in terms of treatment with N16/F15. We apologize that the text was not sufficiently clear and emphatic.

COMMENT 5b: In the same line, upon overexpression of Tsg101 C73A, which should function as a *trans* dominant-negative mutant, N16 should lose its negative impact towards Gag release.

RESPONSE: Tsg101 expression is highly regulated (Lu and Cohen, 2003). Nevertheless, we tried to test the effect of Cys73A over-expression and had no success. Without depletion of the endogenous protein, we were unable to achieve sufficient expression of adventitious Tsg101 to demonstrate Cys73A impact on N16 treatment. We have since approached the question in a depletion-replacement experiment as recommended by Reviewer-1. The results are shown in *Supplementary Figure 10* where, as predicted, budding was found resistant to N16 in cells where Tsg101 has been replacement with C73A while remaining susceptible in cells where replacement was with WT Tsg101.

COMMENT 6: Another control for the specificity of the effect would be to add reagents that control the oxidative state in the extracellular and intracellular milieu. For instance, the addition of antioxidative reagents like glutathione should overrule the N16 effect.

RESPONSE: N-acetyl Cysteine (NAC), another potent antioxidant, was employed in place of glutathione. Testing with the Roche WST-1 reagent showed that 293T cells were robust up to at least 4 mM NAC; we used 3 mM for the experiment. As the Reviewer predicted, the addition of the antioxidative reagent suppressed the N16 inhibitory effect. VL P production and intracellular Gag accumulation were improved in parallel. This information is included in the revised manuscript as *Supplementary Figure S9*.

COMMENT 7: Immune recognition of HIV positive cells are mainly controlled in vivo by Gag-derived MHC I epitopes. Beside the antiretroviral effect, treatment with N16/F15 should also deploy a very interesting side-effect: augmentation of the entry of Gag into the ubiquitin, proteasome and MHC-I pathway. Therefore, the TCD8 response to Gag should be improved, which might be of interest, not only for vaccine development but also for immune control of HIV+ cells during ART. Therefore, it would be a beautiful add-on to the importance of the general finding if the authors could show that N16/F15 increase the MHC I presentation of Gag derived epitopes.

RESPONSE: We consider this suggestion outside the scope of the current study. However, we agree that the hypothesis is of interest.

COMMENT 8: It is well established that proteasome inhibitors exhibit an antiretroviral activity by causing accumulation of polyubiquitinated Gag proteins. With this in mind, the anti-Tsg101 drugs should have a synergistic effect with proteasome inhibitors in terms of virus budding and replication capacity. It would also be nice if the authors could control this aspect.

RESPONSE: We found that the addition of MG132 to cells for just 1 hour prior to disrupting them for lysate preparation (longer is inhibitory) revealed that Gag proteins saved from degradation exhibited a gel migration pattern consistent with modification by polyubiquitination. This effect appeared to be N16 dose-dependent, consistent with the Reviewer's suggestion that the effect of the anti-Tsg101 drug and MG132 might indeed be linked in some manner. Unfortunately, the results were inconsistent, difficult to reproduce, and therefore abandoned as informative to this study at this time.

COMMENT 9a: Several p6 mutants have been described to exhibit an increased Gag polyubiquitination independently of L-domains. One of them, the S40F, has been intensively characterized by the authors. Those mutants, in particular since they have almost normal virus release, should not, or at least less respond to N16/F15 treatment.

RESPONSE: Our results do not support this prediction. As the properties of the S40F mutant are dependent on Alix binding (Watanabe 2013) and the Alix-mediated rescue of P7L budding was as N16 sensitive as WT Gag, we would expect S40F release to also exhibit sensitivity comparable to WT. We performed the experiment and confirmed that the S40F mutation was as sensitive to N16 as the WT parent (*not shown*).

COMMENT 9b: Some of these control experiments should be conducted in order to validate the very interesting finding described in this manuscript.

RESPONSE: Essentially all of the Reviewer's suggestions were attempted (with the exception of the suggestions pertaining to E1 thermosensitive cell lines and MHC-I presentation for the indicated reasons.)

Minor points:

COMMENT 1: [line 39] At least once, the abbreviation UEV should be written-out.

RESPONSE: In the revised manuscript, the abbreviation is written out in the Abstract and in the second paragraph in the Introduction.

COMMENT 2: [Fig.1a, right, top] The electron microscopy pictures only show small sections of budding virions. Why do the authors not show a larger section to demonstrate differences of mock and N16 treated cells or GagP7L? In general, almost any stage of virus morphogenesis can be seen even for wt virus.

RESPONSE: This panel has been expanded in the revised manuscript to include larger sections of budding virions and a quantitative analysis.

COMMENT 3: [line 175-177] The authors state that Gag and Tsg101 co-localize in small puncta <10 nm as measured by confocal microscopy. Since the microscope used by the authors is bound to the Abbe limit, a resolution below 200 nm should not be feasible. The authors should either use STED/PALM super-resolution microscopy or correct their statement.

RESPONSE: The statement has been changed from "*and in smaller puncta <10 nm in diameter*" to "*and in smaller puncta*" (now line 195). We thank the reviewer for pointing out this error.

COMMENT 4: [line 184-187] The authors show that Tsg101 and GagP7L do not co-localize. They should comment in the discussion how Tsg101 can influence GagP7L without interaction.

RESPONSE: The Discussion now comments on how Tsg101 might influence Gag-P7L without interaction. Essentially, P7L might still interact with Tsg101 through the Ub binding pocket OR require a function mediated through that pocket. We also suggest that in the absence of direct interaction, N16 bound to Tsg101 might exert *trans*-dominant negative influence on a function required for Alix-mediated P7L budding.

COMMENT 5: [line 414] misspelling of the word Hoechst.

RESPONSE: The spelling of the word "*Hoechst*" has been corrected. We thank the Reviewer for directing our attention to this error.

Reviewers' comments:

Reviewer #1 (Remarks to the Author):

In this resubmitted MS, Strickland and colleagues report on the effects of a compound that blocks binding between the Tsg101 UEV domain and ubiquitin. They report that this compound disrupts an early step in HIV-1 budding. The authors argue that their data reveal a novel function of Tsg101 as a chaperone that permits bud initiation.

The authors have addressed some of the issues raised in the initial round of review, and have back-tracked from their earlier claim that Tsg101 functions to stabilize Gag. Although the compounds reported here are interesting, the claim that Tsg101 serves a chaperone function in early bud formation is not well supported by either the extensive literature on Tsg101 and HIV budding, or by the data in the paper itself. Specific comments follow.

1. If Tsg101 binding to Ub played a role early in HIV budding and Tsg101 binding to PTAP played the well-established late role in budding, one would expect that Tsg101 depletion would block the first step in the budding pathway at which Tsg101 was required, i.e., early in the budding pathway, according to the authors' model. This is not what one sees: numerous studies have demonstrated that Tsg101 depletion induces a late block, similar to that imposed by p6 deletion or PTAP mutation. The authors' model is thus inconsistent with a large body of literature.
2. Line 142. The authors state that treatment of P7L with N16 blocks the residual particle release that occurs with this PTAP domain mutant. However, nowhere do I find the data showing this. Fig. 1C contains some particle morphogenesis results showing a small increase in "early" vs. "tethered" budding structures when P7L-expressing cells are treated with N16. It is not clear whether this effect is statistically significant, and the authors do not perform the more quantitative virus release assays with P7L that would allow them to reach the above-mentioned conclusion.
3. I remain concerned about toxicity of N16. The authors argue against the observed effects being attributable to "irreversible toxicity" (Fig. S2) but this does not exclude toxicity. In Fig. S5, a striking and dose-dependent reduction in EGF levels is observed upon N16 treatment, suggesting loss of cell viability.

Reviewer #2 (Remarks to the Author):

I have received the revised version of the manuscript entitled „Tsg101 Chaperone Function Revealed by Novel HIV-1 Assembly Inhibitors“ by Dr. Carter et al. I am satisfied with all revisions and my comments were addressed properly.

Reviewer #3 (Remarks to the Author):

In the revised version of their manuscript the authors convincingly dealt with all questions raised by the reviewer, which clearly increased the quality and significance of the quintessence of their finding. Particularly, in response to our request, the destabilizing effect of the Tsg101 inhibitor N16 on Gag is now removed and declared to be rather as an off target effect specifically occurring in 293T-cells.

The revised manuscript is now acceptable for publication.

Comments of Reviewer 1:

In this resubmitted MS, Strickland and colleagues report on the effects of a compound that blocks binding between the Tsg101 UEV domain and ubiquitin. They report that this compound disrupts an early step in HIV-1 budding. The authors argue that their data reveal a novel function of Tsg101 as a chaperone that permits bud initiation.

The authors have addressed some of the issues raised in the initial round of review, and have backtracked from their earlier claim that Tsg101 functions to stabilize Gag. Although the compounds reported here are interesting, the claim that Tsg101 serves a chaperone function in early bud formation is not well supported by either the extensive literature on Tsg101 and HIV budding, or by the data in the paper itself. Specific comments follow.

1. If Tsg101 binding to Ub played a role early in HIV budding and Tsg101 binding to PTAP played the well-established late role in budding, one would expect that Tsg101 depletion would block the first step in the budding pathway at which Tsg101 was required, i.e., early in the budding pathway, according to the authors' model. This is not what one sees: numerous studies have demonstrated that Tsg101 depletion induces a late block, similar to that imposed by p6 deletion or PTAP mutation. The authors' model is thus inconsistent with a large body of literature.

Our Response: The Reviewer makes the expectation of identical outcome for two conditions under which budding occurs. There is no *a priori* basis for such an expectation. Under conditions of Tsg101 depletion or PTAP mutation, based on a body of documented findings, budding is directed by a redundant pathway that is Alix-driven. We show on Fig 4 that despite N16-mediated disruption of Ub binding UEV-PTAP binding proceeds unperturbed. Therefore, under conditions of N16 treatment, budding is Tsg101-PTAP-driven. It is well recognized that there are differences between the two budding pathways in terms of other cellular proteins engaged (Votteler & Sundquist, 14(3):232-41, 2013) and the structure of their bud products (Carlson et al Cell Host & Microbe 4, 592-599, 2008). Therefore, there is no inconsistency in finding differing manifestations of budding arrest under Tsg101 depletion (where budding is mediated redundantly by Alix to a late stage) and under N16 perturbed Tsg101-PTAP-driven budding, which stalls budding early.

In the submitted revision#2, this point is now explicitly made in lines 138-146: Given that PTAP mutation, as well as Tsg101 depletion, arrest budding at the 'Tethered' stage, an intuitive expectation is for the N16 arrest phenotype to be the same. However, there is no *a priori* basis for such an expectation as, under conditions of Tsg101 depletion or PTAP mutation, budding is driven by a redundant pathway directed by Alix²³⁻²⁷. As we will show below (Fig. 4), N16 disrupts the Ub binding pocket but leaves the PTAP-binding pocket unperturbed; hence, under conditions of N16 treatment, budding is still Tsg101-PTAP-driven. It is well-recognized that these two budding pathways differ in terms of other cellular proteins engaged²⁸ and the structure of their bud products²⁹.

2. Line 142. The authors state that treatment of P7L with N16 blocks the residual particle release that occurs with this PTAP domain mutant. However, nowhere do I find the data showing this. Fig. 1C contains some particle morphogenesis results showing a small increase in "early" vs. "tethered" budding structures when P7L-expressing cells are treated with N16. It is not clear whether this effect is

statistically significant, and the authors do not perform the more quantitative virus release assays with P7L that would allow them to reach the above-mentioned conclusion.

Our Response: The current submission now contains both pieces of supporting information: (i) The statistical significance for the change in the stage at which budding is arrested following N16 treatment of P7L has a P value in the Chi Square test of 0.0001 and is therefore considered to be significant. (ii) Isolation of P7L NL4-3 virus particles followed by Western analysis and quantitative virus release assay is provided to support the conclusion (new *Supplementary Figure 3*). We agree with the Reviewer that the quantitative aspect of this virus release assay is a necessary addition to the morphological examination by EM.

In the submitted revision#2, the reader is directed to *Supplementary Figure 3* in lines 150-153: As indicated by Western analysis, treatment of P7L with N16 blocked the residual particle release that the mutation permits (mediated by Alix, an ESCRT adaptor³²) and EM analysis indicated that the predominant form shifted from 'Tethered' to 'Early' (*Supplementary Figure S3*).

3. I remain concerned about toxicity of N16. The authors argue against the observed effects being attributable to "irreversible toxicity" (Fig. S2) but this does not exclude toxicity. In Fig. S5, a striking and dose-dependent reduction in EGF levels is observed upon N16 treatment, suggesting loss of cell viability.

Our Response: We remind the reviewer that N16 is part of a family of closely related compounds that have all been approved for human consumption since ~ the year 2000 in most cases. N16 itself is undergoing clinical trials; note, in addition that we are seeing an effect at a level that is within 2-fold of the concentration in clinical trial patients' plasma. To date, we have assessed N16 toxicity over a concentration range of zero to 150 micromolar using a number of different assays:

(1) Formulations for assessment of tissue culture cell metabolic activity marketed by Promega (the *CellTiter 96 AQueus One Solution cell proliferation assay system*) and Roche (the WST-1 reagent). These assay systems contain a tetrazolium compound and an electron coupling reagent. In both cases, the dye formed directly correlates to the number of metabolically active cells in the culture. To date, 6 cell lines (293T, HeLa, Cos-1, Vero, Jurkat and MT-4) have been examined. The response shown in *Supplementary Figure 6a* is representative.

(2) We have demonstrated that the inhibitory effect of N16 on virus production are neutralized by N-Acetyl Cysteine, (*Supplementary Figure 10*) supporting the specificity of its action in targeting of Cys.

(3) We have shown that N16 inhibition is suppressed by replacing endogenous Tsg101 with Tsg101-C73A, a mutant lacking the N16 target. This demonstrates that N16 targets the specific Cys residue predicted by our NMR structure and demonstrated by our confocal microscopy analysis and again supports the specificity of N16 action (*Supplementary Figure 11*).

(4) We demonstrated that the inhibitory activity of N16 on virus production was manifested only when the drug was administered in proximity to the onset of Gag gene transfection or virus infection (*i.e.*, T = -6 to +6 hr; *Figure 1*). Thus, no toxicity that impacted virus replication was apparent following N16 exposure after this period even though the drug was present at the same concentration and duration.

(5) We have demonstrated that supplying N16 daily over a 15 day period did not prevent cells from replicating and producing virus following removal of the drug (*Supplementary Figure S2*).

In the submitted revision#2, the issue regarding N16 toxicity is now more deliberately summarized in the Discussion, lines 400-420: In this study, we identified a class of drugs through a screen (*cf.*, Fig S1) and showed by NMR analysis (*i*) that its activated form covalently binds the Cys73 residue of the UEV domain of Tsg101 and (*ii*) that this resulted in disruption of UEV Ub binding but not PTAP binding activity (*cf.*, Fig 4, 5, S7, S8, S9). As inhibitors of HIV-1 assembly, the drugs' action appears to be specifically targeted: (1) Inhibitory effects on Gag assembly were observed at concentrations well below drug toxicity as assessed using tests based on cell metabolic activity (*cf.*, Methods and Materials; Fig S6a). (2) Inhibition of virus production by the drug was neutralized by N-Acetyl Cysteine (*cf.*, Fig S10) supporting the specificity of drug targeting to a Cys residue in the Tsg101-UEV domain (*cf.*, Fig 5). (3) Inhibition of virus production was suppressed by replacing endogenous Tsg101 with Tsg101-C73A, a Tsg101 mutant lacking the residue with which the drug forms a covalent adduct. In contrast, replacement with WT Tsg101 did not suppress the inhibition (*cf.*, Fig S11). (4) Inhibition of virus production (*cf.*, Fig 1) and perturbations in trafficking (*cf.*, Fig 2) and bud formation (*cf.*, Fig 1) were manifested only when the drug was administered early (*i.e.*, between 6 hrs *pre*- and 5 hrs *post*-transfection of DNA encoding Gag and examined 24 hrs later. When added at 24 hrs *post*-transfection and examined 24 hrs later, Gag assembly was not perturbed even though the drug concentration and exposure time were the same. Thus, based on their specificity for targeting HIV-1 assembly and lack of cellular toxicity, the drugs are useful tools for investigating the contribution of the Ub binding activity of Tsg101 in its various biological roles.

(6) Regarding the *reduction in EGFR levels observed upon N16 treatment*: As we demonstrated, neither ligand-induced EGFR down-regulation nor cytokinesis showed detectable reduction. In contrast, a reduction in EGFR levels is apparent in the absence of ligand addition upon N16 treatment, as the reviewer points out (*Supplementary Figure 6b, lanes 3, 5, 7 compared to lane 1*). All 3 activities, constitutive EGFR trafficking, ligand-induced EGFR down-regulation and participation in abscission during cytokinesis, are reported functions of Tsg101 (Rush & Ceresa *Mol Cell Endocrinol*, 381:188-97, 2013; Lu et al *PNAS U S A*, 100:7626-31,2003; Carlton & Martin-Serrano *Science*. 2007 316:1908-12, 2007; Morita et al *EMBO J*. 26:4215-27, 2007). Rather than being indicative of loss of cell viability as the reviewer suggests, we interpret this finding to suggest that the Tsg101 Ub-binding function that is disrupted by N16 normally participates in this particular Tsg101 function. We thank the reviewer for calling attention to this point.

In the submitted revision#2, the point is brought out in lines 262-269: The down-regulation function remained unimpaired at concentrations well above 50 μ M. The results indicated that these well-established cell-directed Tsg101 functions were resistant to N16 at the concentration to which virus production was susceptible. In contrast, the steady state level of unliganded EGFR (-EGF lanes) appears to be N16 sensitive (compare DMSO control/lane 1 and +50 μ M N16/lane 3). Intriguingly, constitutive recycling of unliganded EGFR is a recent addition to the list of cellular functions of Tsg101: Depletion of Tsg101 caused unliganded EGFR to traffic to lysosomes⁴⁴.